# Cerebral microstructural alterations in Post-COVID-condition are related to cognitive impairment, olfactory dysfunction and fatigue

Jonas A. Hosp [1,9] ✉, Marco Reisert[2,3,9], Andrea Dressing[1,4], Veronika Götz[5], Elias Kellner [2], Hansjörg Mast[6], Susan Arndt[7], Cornelius F. Waller [8], Dirk Wagner[5], Siegbert Rieg[5], Horst Urbach[6], Cornelius Weiller [1], Nils Schröter [1,9] & Alexander Rau [6,9]

After contracting COVID-19, a substantial number of individuals develop a Post-COVID-Condition, marked by neurologic symptoms such as cognitive deficits, olfactory dysfunction, and fatigue. Despite this, biomarkers and pathophysiological understandings of this condition remain limited. Employing magnetic resonance imaging, we conduct a comparative analysis of cerebral microstructure among patients with Post-COVID-Condition, healthy controls, and individuals that contracted COVID-19 without long-term symptoms. We reveal widespread alterations in cerebral microstructure, attributed to a shift in volume from neuronal compartments to free fluid, associated with the severity of the initial infection. Correlating these alterations with cognition, olfaction, and fatigue unveils distinct affected networks, which are in close anatomical-functional relationship with the respective symptoms.

The long-term consequences of the Corona Virus Disease 2019 (COVID-19) not only include shortness of breath or palpitations but also encompass a multitude of neurological and neurocognitive symptoms such as fatigue, persistent impairment of olfaction, and deficits in attention and memory[1,2]. Accordingly, the World Health Organization (WHO) published the following diagnostic criteria to define Post-COVID-Condition (PCC): History of probable or confirmed SARS-CoV-2 (Severe acute respiratory syndrome coronavirus type 2)

infection; at least one symptom (e.g., fatigue, shortness of breath, or cognitive dysfunction) with relevant impact on everyday functioning; symptom persistence for at least two months; and a delay of at least 3 months between onset of acute COVID-19 and diagnosis[3]. Among the PCC-associated symptoms, neurological/neurocognitive deficits occur frequently and can substantially contribute to disease burden[2]. Given that approximately 10 to 25% of COVID-19 survivors develop PCC[1,2] and 8% (population-based data)[4] to 39% (selected cohort)[5] report

[1]Department of Neurology and Clinical Neuroscience, Medical Center – University of Freiburg, Faculty of Medicine, University of Freiburg, Freiburg, Germany. [2]Department of Diagnostic and Interventional Radiology, Medical Center – University of Freiburg, Faculty of Medicine, University of Freiburg, Freiburg, Germany. [3]Department of Stereotactic and Functional Neurosurgery, Medical Center – University of Freiburg, Faculty of Medicine, University of Freiburg, Freiburg, Germany. [4]Freiburg Brain Imaging Center, Medical Center – University of Freiburg, Faculty of Medicine, University of Freiburg, Freiburg, Germany. [5]Department of Internal Medicine II, Medical Center – University of Freiburg, Faculty of Medicine, University of Freiburg, Freiburg, Germany. [6]Department of Neuroradiology, Medical Center – University of Freiburg, Faculty of Medicine, University of Freiburg, Freiburg, Germany. [7]Department of Otorhinolaryngology - Head and Neck Surgery, Medical Center - University of Freiburg, Faculty of Medicine, University of Freiburg, Freiburg, Germany. [8]Department of Internal Medicine I, Medical Center - University of Freiburg, Faculty of Medicine, University of Freiburg, Freiburg, Germany. [9]These authors contributed equally: Jonas A. Hosp, Marco Reisert, Nils Schröter, Alexander Rau. ✉e-mail: jonas.hosp@uniklinik-freiburg.de

detrimental effects on work-life, this disease is likely to have significant socioeconomic consequences.

Nonetheless, the pathophysiology of neurocognitive deficits in PCC remains poorly understood and evidence for biomarkers is currently lacking. Recently, a large, longitudinal MRI-based study from the UK Biobank compared pre- and post-COVID-19 brain scans four to five months after infection[6]. Here, increased diffusion indices were observed within the limbic regions (e.g., anterior cingulate, hippocampal, parahippocampal, and orbito-frontal cortex) and striatum, whereas cortical thickness was reduced in orbitofrontal and parahippocampal regions. Furthermore, there was a greater reduction in global brain size in the COVID-19 group compared to controls. As a similar pattern of cortical atrophy was also detected in a cohort of PCC-patients[7], we hypothesized that these marked changes in brain (micro-) structure could form the pathophysiological basis of PCC-related symptoms such as cognitive impairment, disturbed olfaction or fatigue.

To investigate this hypothesis, we applied multi-compartment diffusion microstructure imaging (DMI) - a powerful tool for detecting even subtle changes within the brain's meso-/microstructure. DMI is based on multi-shell diffusion protocols and allows different anatomical compartments to be segregated based on their particular diffusion properties[8-10]. This technique has allowed the detection of widespread vasogenic white matter edema in hospitalized subacute COVID-19 patients who had developed neurological symptoms[11].

In this study, we analyze cortical morphometry and DMI metrics in a prospective monocentric cohort of patients diagnosed with PCC according to the WHO criteria[3], participants that had contracted COVID-19 without lasting symptoms, as well as healthy controls that had never suffered from COVID-19. We show that COVID-19 leads to microstructural changes in the brain, which differ between participants with and without PCC symptoms. Correlations between imaging parameters and clinical symptoms reveal affection of distinct cerebral networks related to cognitive or olfactory impairment, and fatigue.

## Results

### Demographic and clinical characteristics

We present the MRI data from a cohort of 89 patients (age median 49; IQR [23] years; range: 19 to 72 years; 55 females) who fulfilled the WHO diagnostic criteria for Post-COVID-Condition (PCC group). Demographics and patient characteristics are listed in Tables 1 and 2, as well as Supplementary Data 1 and 2. Although neurological examinations did not reveal any focal deficits, patients complained about impaired attention and memory (100%), fatigue (96%), impaired multi-tasking ability (97%), word-finding difficulties (89%), and disrupted olfaction (50%). Due to these symptoms, 36 patients (40%) were unable to work and 11 (13%) had to reduce their workload. Seventy-six patients (85%) had a mild course of acute COVID-19 without hospitalization. Regarding MoCA performance, 36 patients (41%) performed below the cut-off score for cognitive impairment[12]. Amongst the MoCA sub-scores, the most pronounced deficits were present in the domain memory, with only slight deficits observed in the domains visuoconstruction and execution. Olfactory performance was impaired in 66 (74%). Using a self-rating questionnaire (WEIMuS) 65 out of 83 patients (78%) revealed overall symptoms of fatigue. On a subscore level, 76%

## Table 1 | Demographics and comorbidities of study participants

| Demographic data | Post-COVID- Condition (PCC; n = 89) n (%) or median [IQR]; range | Unimpaired Post-COVID (UPC; n = 38) (%) or median [IQR]; range | Healthy Controls (HC; n = 46) (%) or median [IQR]; range | P value |
|---|---|---|---|---|
| Age (years) | 49 [23]; 19 to 72 | 42 [24]; 25 to 62 | 44 [31]; 21 to 80 | 0.2192[a] |
| Sex (male/female) | 34 (38)/55 (62) | 13 (34)/25 (66) | 23 (50)/23 (50) | 0.281[b] |
| Δ positive PCR - cMRI (days) | 254 [209]; 90 to 710 | 227 [292]; 145 to 943 | NA | 0.626[a] |
| Comorbidities | n (%) | n (%) | n (%) | P value[c] |
| Achalasia | 0 (0%) | 0 (0%) | 1 (2%) | 0.485 |
| Adipositas | 12 (14%) | 3 (8%) | 0 (0%) | 0.017 |
| Arterial hypertension | 19 (21%) | 4 (10%) | 0 (0%) | <0.001 |
| Asthma/COPD | 8 (9%) | 1 (3%) | 1 (2%) | 0.238 |
| Atrial fibrillation | 1 (1%) | 0 (0%) | 0 (0%) | 0.999 |
| Chronic kidney failure | 1 (1%) | 0 (0%) | 0 (0%) | 0.999 |
| Coronary heart disease | 3 (3%) | 1 (3%) | 1 (2%) | 0.999 |
| Deep vein thrombosis | 0 (0%) | 0 (0%) | 1 (2%) | 0.485 |
| Diabetes | 5 (6%) | 1 (3%) | 0 (0%) | 0.256 |
| Gout | 0 (0%) | 0 (0%) | 1 (2%) | 0.485 |
| History of depression | 9 (10%) | 2 (5%) | 0 (0%) | 0.06 |
| History of ischemic stroke | 2 (2%) | 0 (0%) | 0 (0%) | 0.725 |
| Fibromyalgia | 0 (0%) | 0 (0%) | 1 (2%) | 0.485 |
| Hyperthyreosis | 0 (0%) | 0 (0%) | 1 (2%) | 0.485 |
| Hypothyreosis | 9 (8%) | 6 (16%) | 3 (6%) | 0.366 |
| Malignancy | 3 (3%) | 0 (0%) | 5 (11%) | 0.051 |
| Migraine | 10 (11%) | 1 (3%) | 0 (0%) | 0.023 |
| Obstructive sleep apnoea | 7 (8%) | 0 (0%) | 0 (0%) | 0.041 |
| Peripheral arterial occlusive disease | 1 (1%) | 0 (0%) | 0 (0%) | 0.999 |
| Restless legs syndrome | 2 (2%) | 0 (0%) | 0 (0%) | 0.725 |
| Rheumatoid arthritis | 0 (0%) | 0 (0%) | 1 (2%) | 0.485 |
| Spinal disc herniation | 0 (0%) | 0 (0%) | 1 (2%) | 0.485 |

An extended version of this table with further statistical details can be found in the Supplementary Data 1.
[a]Kruskal–Wallis-Test.
[b]X²-Test.
[c]Fisher's Exact Test.

**Table 2 | Clinical characteristics of study participants**

| | Post-COVID-Condition (PCC; $n=89$) | Unimpaired Post-COVID (UPC; $n=38$) | |
|---|---|---|---|
| | median [IQR]; range; n outside of norm (%) | median [IQR]; range; n outside of norm (%) | P value |
| Disease severity score[a] | | | |
| Mild course of COVID-19 | 76 (85%) | 37 (97%) | 0.048[c] |
| 1 | 42 (47%) | 25 (65%) | |
| 2 | 34 (38%) | 12 (32%) | |
| Severe course of COVID-19 | 13 (15%) | 1 (3%) | 0.048[c] |
| 3 | 11 (13%) | 1 (3%) | |
| 4 | 2 (2%) | 0 (0%) | |
| Grading of current disability[b] | | | |
| 0 | 0 (0%) | 38 (100%) | |
| 1 | 42 (47%) | 0 (0%) | |
| 2 | 11 (13%) | 0 (0%) | |
| 3 | 36 (40%) | 0 (0%) | |
| Current neurological symptoms | | | |
| Disturbed olfaction/gust | | | |
| initially | 66 (74%) | 12 (31%) | |
| currently | 44 (50%) | 0 (0%) | |
| Impaired attention | 89 (100%) | 0 (0%) | |
| Memory impairment | 89 (100%) | 0 (0%) | |
| Impaired multi-tasking | 86 (97%) | 0 (0%) | |
| Word-finding difficulties | 79 (89%) | 0 (0%) | |
| Fatigue | 85 (96%) | 0 (0%) | |
| Clinical readouts | | | |
| MoCA sum score (corrected for years of education; norm ≥ 26/30) | 26 [4]; 18 to 30; 36 (41%) | 27 [3]; 23 to 30; 7 (18%) | 0.003[d] |
| MoCA domain scores | | | |
| - Orientation (max. 6 points) | 6 [0]; 5 to 6 | 6 [0]; 5 to 6 | |
| - Attention (max. 6 points) | 6 [0]; 3 to 6 | 6 [0]; 4 to 6 | |
| - Language (max. 5 points) | 5 [0]; 3 to 5 | 5 [1]; 3 to 5 | |
| - Executive (max. 4 points) | 3 [1]; 2 to 4 | 4 [0]; 3 to 4 | |
| - Visuoconstructive (max. 4 points) | 4 [1]; 1 to 4 | 4 [0]; 3 to 4 | |
| - Memory (max. 5 points) | 3 [3]; 0 to 5 | 4 [3]; 0 to 5 | |
| Correct perception of smell (norm ≥11/12) | 9 [4]; 0 to 12; 66 (74%) | 11 [2]; 7 to 12; 16 (42%) | <0.001[d] |
| Würzburg Fatigue Inventory in Multiple Sclerosis score (WEIMuS; norm: ≤33/68)[e] | 43 [17]; 4 to 65; 65 (78%) | 7 [18]; 0 to 48; 3 (8%) | <0.001[d] |
| - cognitive fatigue (norm: ≤17/36) | 22 [10]; 0 to 35; 63 (76%) | 4 [10]; 0 to 24; 4 (11%) | <0.001[d] |
| - physical fatigue (norm: ≤16/32) | 22 [9]; 4 to 32; 64 (77%) | 3 [10]; 0 to 24; 1 (3%) | <0.001[d] |
| Geriatric Depression Scale (GDS-15; norm: ≤7/15)[e] | 4 [5]; 1 to 13; 16 (19%) | 1 [2]; 0 to 9; 1 (3%) | <0.001[d] |

An extended version of this table with further statistical details can be found in the Supplementary Data 2.

[a]Disease severity score: (1) no pneumonia; (2) pneumonia, outpatient treatment; (3) pneumonia, inpatient treatment; (4) ARDS, endotracheal ventilation at ICU.

[b]Grading of current disability: (0) no relevant restrictions; (1) relevant restrictions of daily life activities but able to work; (2) reduction of work quota necessary; (3) inability to work and restriction of daily life activities.

[c]X² test.

[d]Mann–Whitney-U test.

[e]data available only for $n=83$ patients.

($n=63$) were above the cut-off for mental fatigue and 77% ($n=64$) above that for physical fatigue. The GDS-15 indicated no relevant level of depression in the PCC cohort at the group level, although 16 patients (19%) did exceed the cut-off value. Cross-comparison of the parameters current disability, disease severity, MoCA-performance, WEIMuS, olfactory performance, and GDS-15, only revealed a significant association between GDS-15 and WEIMuS ($P=0.012$), after correcting for multiple comparisons.

Furthermore, 38 participants (age median 42; IQR [24] years; range: 25–62 years; 25 females) with a previous COVID-19 infection but without lasting subjective impairment (i.e., Unimpaired Post-COVID-group; UPC) were enrolled in the study. Detailed demographic and clinical information is provided in Tables 1 and 2. Statistical comparison of the UPC and PCC groups revealed no significant differences in terms of age, sex, and the delay between positive PCR and imaging (Table 1). Interestingly, a severe course of acute COVID-19 was more frequent in the PCC group ($X^2$, $P=0.048$). Regarding clinical readouts, UPC participants performed significantly better in cognitive testing (Mann–Whitney-U, $P=0.003$) and olfactory testing, (Mann–Whitney-U, $P<0.001$) and were significantly less affected by depression or fatigue (both Mann–Whitney-U, $P<0.001$). Analysis of comorbidity occurrence revealed significant differences between groups (PCC, UPC, and HNC) in terms of adipositas, arterial hypertension, migraine, and obstructive sleep apnoea, to the detriment of patients with PCC (see Table 1).

## Evaluation of conventional MRI and cortical morphometric analysis

Mild microangiopathic white matter changes corresponding to Fazekas 1 were present in 6 patients with PCC[13]. In one patient (39-year-old female), a small, primarily gliotic lesion in the right basal ganglia was found. An occipital cortical defect in another patient (66-year-old male) was most likely a post-ischemic lesion. In a third patient (62-year-old male), slight T2 signal elevations were observed bilaterally in the globus pallidum, without any correlation to other MRI sequences. No further structural changes, signs of atrophy or any evidence of inflammation (e.g., leptomeningeal enhancement) were found. Within the UPC group, we found symmetrical hyperintense T2 signals of unknown origin in one patient (39-year-old male) and slight microangiopathic lesions in two patients (both Fazekas 1)[13].

Region-wise FreeSurfer-based[14] comparison of cortical thickness, surface area, and gray-matter volume as well as comparison of total gray-matter volume, (i.e., two-sided ANCOVAs controlling for age and sex to assess intergroup differences between PCC, UPC, and HNC for the aforementioned parameters) yielded no statistically significant differences between groups after FWE-correction for multiple comparisons (Supplementary Data 3 and 4). Thus, there was no evidence of cortical or global gray matter atrophy in either the UPC or PCC group.

## Group comparison of gross changes in DMI parameters in the gray matter

Intergroup comparison of whole-brain gray/white matter DMI parameters was performed using two-tailed ANCOVAs, controlling for age and sex. Bonferroni corrections were performed to account for multiple comparisons. With respect to gray matter (Fig. 1), a significant group effect on V-extra could be detected (df = 168, $P<0.001$; $F=14.999$). Post-hoc tests revealed a significant reduction in V-extra in the PCC group compared to UPC ($P<0.001$, $t=-4.463$, Cohen's $d=0.794$, 95% CI [0.398, 1.190]) and HNC controls ($P<0.001$, $t=-4.407$, Cohen's d = 0.510, 95% CI [0.146, 0.875]), whereas no statistically significant difference was found between the latter groups ($P=0.951$, $t=0.301$, Cohen's d = −0.411, 95% CI [−0.852, 0.029]). For V-CSF, a significant group effect existed (df = 168, $P<0.001$, $F=12.890$),

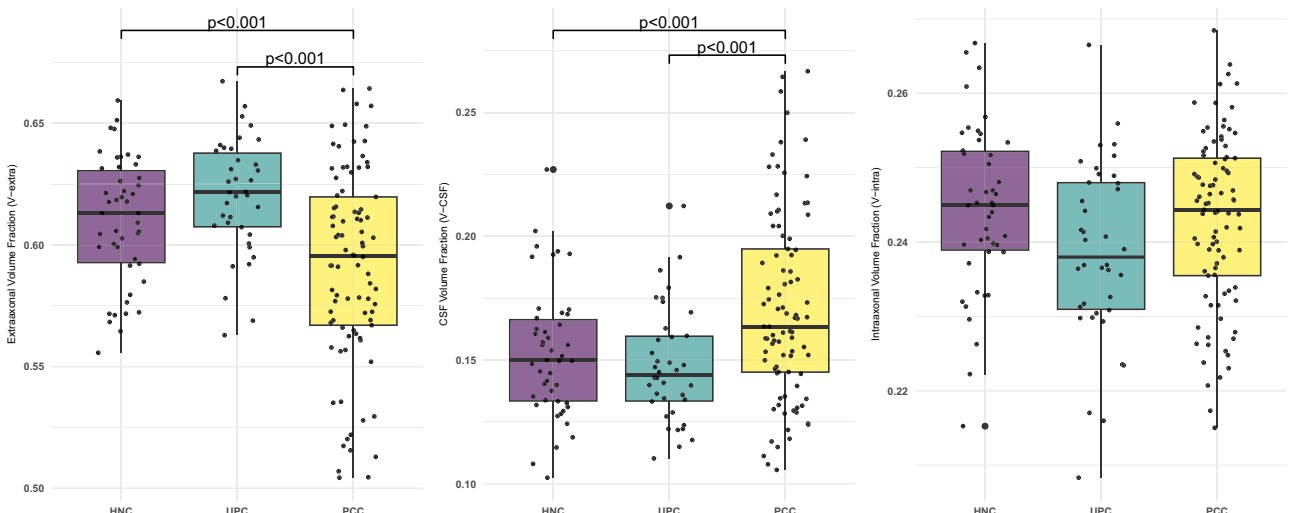

**Fig. 1 | Gray matter DMI parameters indicate microstructural changes between groups.** Box plots show the distribution of microstructural compartments within the entire gray matter. In patients with Post-COVID-Condition (PCC), a significant decrease in the extraneurite volume fraction (V-extra) was accompanied by a significant increase in the free-fluid fraction (V-CSF) when compared to Unimpaired Post-COVID (UPC) patients and the Healthy Non-COVID (HNC) group. Regarding the intraneurite volume fraction (V-intra), no statistically significant differences were present between groups. Group comparisons were performed using two-tailed ANCOVAs, with age and sex as nuisance covariates. The statistic was based on 46 HNC, 38 UPC, and 89 PCC cases. The center lines represent the median, box bounds show interquartile range (IQR), whisker borders indicate 1.5 x IQR. Dots represent individual subject values.

with a consecutive increase in V-CSF in the PCC group compared to UPC ($P = 0.001$, $t = 3.565$, Cohen's d = $-0.688$, 95% CI [$-1.081$, $-0.295$]) and HNC controls ($P < 0.001$, $t = 4.546$, Cohen's d = $-0.538$, 95% CI [$-0.902$, $-0.173$]). No statistically significant difference was found between the UPC and HNC control groups ($P = 0.817$, $t = 0.604$, Cohen's d = $0.204$, 95% CI [$-0.233$, $0.641$]). For V-intra, no statistically significant group effect was detected (df = 168, $P = 0.401$, $F = 2.748$). Furthermore, investigating the white matter DMI parameters in the aforementioned statistical models (i.e., two-sided ANCOVAs controlling for age and sex to assess intergroup differences between PCC, UPC, and HNC for V-extra, V-CSF, and V-intra in whole-brain white-matter), did not yield a statistically significant difference between groups after correction for multiple comparisons (Supplementary Data 5). To address the impact of initial disease severity, we included the disease severity score as a nuisance covariate (together with age and sex) in the ANCOVAs comparing whole-brain parameters between the PCC and UPC groups. While disease severity contributed significantly to the decrease in gray matter V-extra (df = 122, $P < 0.001$, $F = 16.548$) and increase in gray-matter V-CSF (df = 122, $P = 0.003$, $F = 13.876$), the group-effect still remained significant for V-extra (df = 122, $P = 0.008$, $F = 12.106$). In contrast, the delay between positive SARS-CoV-2 PCR and MRI-scan did not significantly contribute to changes in gray-matter DMI parameters between the PCC and UPC groups when added as a nuisance covariate together with age and sex (Supplementary Data 6). Therefore, based on whole-brain metrics, the volume shift from V-extra into V-CSF that was detected in the gray matter tissue was partially - but not exclusively - associated with initial disease severity. Since the patients with Post-COVID-Condition were recruited in two periods, we conducted the comparison of whole-brain gray matter diffusion MRI parameters separately for both periods. Here, comparison with HNC- (Supplementary Fig. 1) and UPC participants (Supplementary Fig. 2) were similar to the pooled analysis. It is also worth noting that this biophysically motivated approach was initially developed for the white matter, which implies a less stable differentiation between V-intra and V-extra in gray matter[9]. Nevertheless, our findings suggest a shift from the membrane-enclosed compartment (V-intra + V-extra) towards V-CSF. As the three-compartment model defines 1 = V-CSF + V-intra + V-extra (i.e., connecting V-extra changes to inverse changes of V-CSF, no

changes in V-intra), we restricted further analyses to the parameter V-extra, not least because we observed the largest effects on this parameter and the neurite fraction V-intra is rather small within the gray matter.

## Spatial distribution and direction of V-extra changes across groups

To display the spatial distribution of changes in V-extra across groups, we performed voxel-wise comparisons (SPM with threshold-free cluster enhancement and FWE-correction for multiple comparisons), using age and sex as nuisance covariates. To further indicate the direction of microstructural alterations (i.e., V-extra increase vs. decrease), we extracted the standardized regression coefficients (beta) of the factor V-extra from regression models. The results of these comparisons are presented in Fig. 2: (1) Pooled PCC and UPC group (i.e., Post-COVID infection) vs. HNC participants: here, reduced V-extra was found to be confined to neocortical gray matter with slight accentuation in the left insular and opercular region and the thalamus. In contrast, increased V-extra is present within the corpus callosum, internal capsule, cerebellum (vermis, cerebellar nuclei, and cortex), mesencephalon, and pons. Mixed effects were noted in mesiotemporal areas. The effects of V-CSF, however, comprised a similar pattern with opposite effect directions indicating opposing compartmental shifts (Fig. 2A). (2) PCC vs. HNC group: significantly different voxels were noted in neocortical gray matter with slight accentuation in the left insular and opercular region and the thalamus; these overlapped with reduced V-extra according to the beta map (Fig. 2B). (3) UPC vs. HNC group: significantly different voxels were particularly present within the infratentorial gray and white matter, with punctum maximum in the mesencephalon and cerebellum (vermis, cerebellar nuclei, and cortex). Regarding supratentorial areas, significantly different voxels were present in the corpus callosum, the internal capsule and mesiotemporal areas. These regions were characterized by an increase in V-extra (Fig. 2C). (4) PCC vs. UPC group: significantly different voxels show a pattern equivalent to the combination of (2) and (3). Regarding the cortical and pulvinar regions, V-extra was more strongly reduced in the PCC than the UPC group. In turn, the increase in V-extra within the supratentorial white matter, mesiotemporal structures, brainstem and cerebellum was stronger in the UPC than PCC group (Fig. 2D).

 

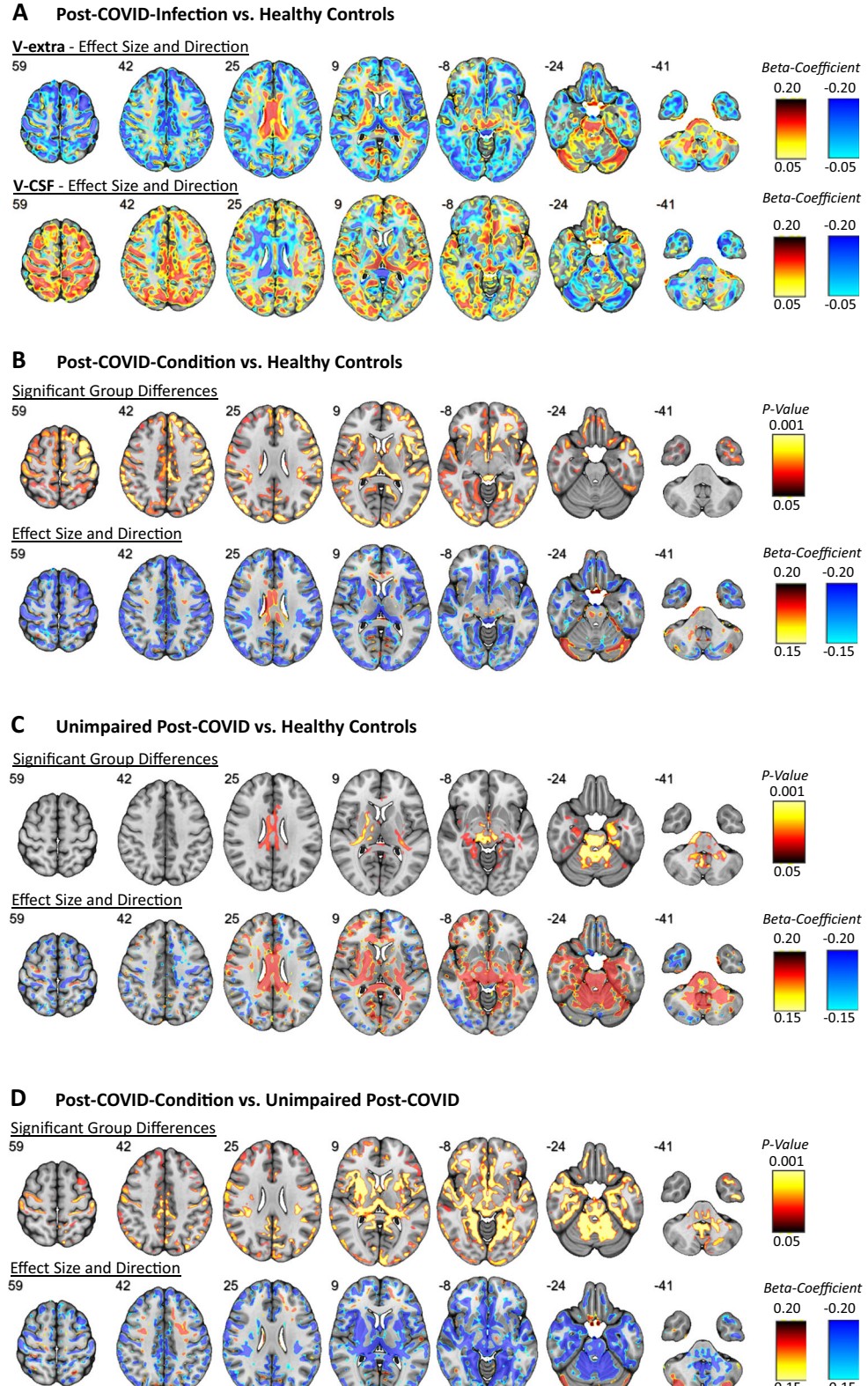

**A   Post-COVID-Infection vs. Healthy Controls**

_V-extra_ - Effect Size and Direction

_V-CSF_ - Effect Size and Direction

**B   Post-COVID-Condition vs. Healthy Controls**

Significant Group Differences

Effect Size and Direction

**C   Unimpaired Post-COVID vs. Healthy Controls**

Significant Group Differences

Effect Size and Direction

**D   Post-COVID-Condition vs. Unimpaired Post-COVID**

Significant Group Differences

Effect Size and Direction

Although disease severity significantly affected the whole gray matter V-extra (see above), adding it into the model as a nuisance covariate did not change the spatial distribution of the results (Supplementary Fig. 3). Likewise, adding the delay between positive SARS-CoV-2 PCR and MRI scan as a nuisance covariate did not explain the spatial distribution of significant V-extra changes between the PCC- and UPC-group (Supplementary Fig. 4). Regarding the two recruitment periods

of patients with Post-COVID-Condition (PCC), we performed voxel-based comparisons of V-extra of patients enrolled in period 1 with the combined collective (i.e., period 1 and 2). Here, adding the PCC-patients enrolled in period 2 did not change spatial patterns of significant effects or effect size/directions expressed by beta-coefficients for both, PCC vs. HNC (Supplementary Fig. 5) and PCC vs. UPC (Supplementary Fig. 6).

**Fig. 2 | Spatial distribution and direction of microstructural changes after COVID-19. A** The standardized regression coefficients of the factors V-extra and V-CSF were extracted from two-tailed linear regression models yielded by voxel-wise comparisons between all participants after COVID-19 infection (i.e., combined PCC and UPC group) and controls without a history of COVID-19 (HNC; with nuisance covariates age and sex), and were superimposed onto a T1w MRI template. Color-coding indicates the coefficient values as a measure of effect size of the factor COVID-19 (hot colors: positive effects vs. cold colors: negative effects). Results of voxel-based two-tailed linear regression models of V-extra after threshold-free cluster enhancement and FWE-correction (top row), and standardized regression coefficients derived from the same model (bottom row) between different groups: **B** Post-COVID-Condition (PCC) vs. Healthy Non-COVID controls

(HNC), **C.** Unimpaired-Post-COVID participants (UPC) vs. Healthy Non-COVID controls (HNC), and (**D**). Post-COVID-Condition (PCC) vs. Unimpaired-Post-COVID (UPC). Voxels with significantly different V-extra were indicated by hot shading and superimposed onto a T1w MRI template (top rows). Two-tailed $P$ values were corrected for multiple comparisons across voxels using the family-wise error rate (FWE), with age and sex as nuisance covariates. Color-coding indicates the coefficient values as a measure of effect size of the factor COVID-19 (hot colors: positive effects vs. cold colors: negative effects; bottom rows). Radiological orientation: left side of the image corresponds to the patient's right; numbers denote the axial ($z$) position in millimeters. The statistic was based on 46 HNC, 38 UPC, and 89 PCC cases.

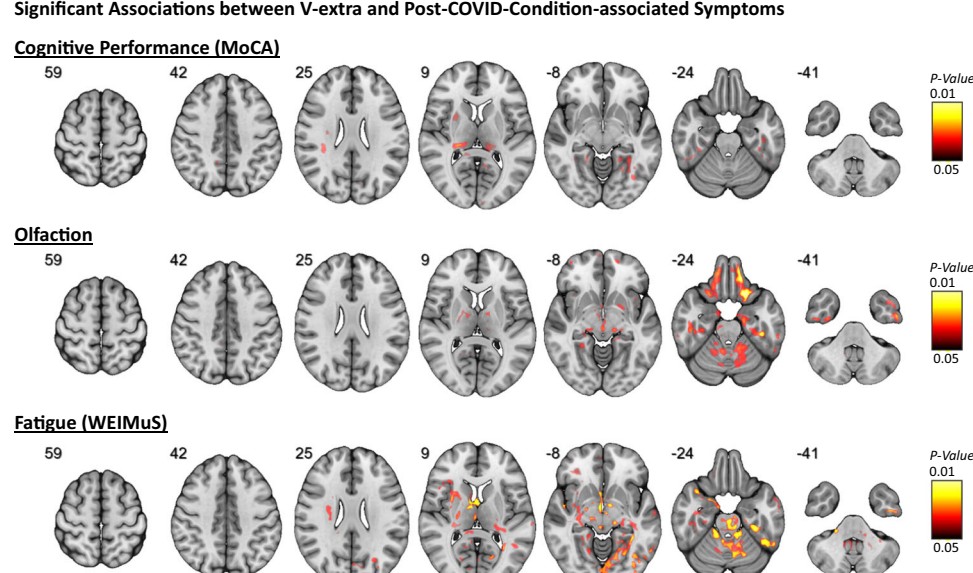

**Significant Associations between V-extra and Post-COVID-Condition-associated Symptoms**

**Cognitive Performance (MoCA)**

**Olfaction**

**Fatigue (WEIMuS)**

**Fig. 3 | Significant association between V-extra and symptoms associated with Post-COVID-Condition.** To relate clinical outcomes to V-extra in patients with a previous COVID-19 infection (i.e., combined PCC- and UPC-group), voxel-based two-tailed linear regression models were employed, with V-extra as a dependent variable, covariates age and sex, and threshold-free cluster enhancement. Two-tailed $P$ values were corrected for multiple comparisons across voxels using the

family-wise error rate (FWE). Voxels with significantly different V-extra are indicated by hot shading and superimposed onto a T1w MRI template. Radiological orientation, i.e., left side of the image corresponds to the right side of the patient's body; numbers denote the axial ($z$) position in millimeters. This statistic was based on $n = 127$ patients with previous COVID-19 infection.

## Clinical measures are associated with changes in V-extra in specific networks

To search for microstructural correlates of PCC-associated symptoms, we performed voxel-based analyses with V-extra as the dependent variable and threshold-free cluster enhancement, with age and sex as nuisance covariates. FWE-correction was performed to control for multiple comparisons ($P < 0.05$). Anatomical allocations were performed using standard atlases[15,16]. No significant voxel association was observed for depression, as measured by the GDS-15. Regarding cognitive performance, a significant inverse association with reduced V-extra was found within the thalamus and mesiotemporal regions (including the hippocampus and parahippocampal gyrus), where this effect was more pronounced in the left hemisphere (i.e., reduced V-extra is associated with low performance; Fig. 3 top; see Supplementary Data 7 for anatomical allocations). Voxel-wise associations with olfactory performance revealed an inverse correlation with reduced V-extra in orbitofrontal cortical areas, mesiotemporal cortex (including inferior-temporal, enthorinal, and parahippocampal cortex, amygdala, and hippocampus), thalamus and cerebellum (i.e., reduced V-extra is associated with low performance; Fig. 3 middle; see Supplementary Data 7 for anatomical allocations). With respect to fatigue, reduced V-extra was found in left temporo-occipital structures (including gyrus lingualis, fusiformis and parahippocampalis), right-

sided putamen, periopercular areas (pars orbitalis, insula and opercular cortex), ventral diencephalon, brainstem and cerebellum (Fig. 3 bottom). To obtain better coverage of brainstem nuclei and diencephalic structures, additional atlases were used for defining anatomical locations[17,18]; Supplementary Data 7. Accordingly, an involvement of the aminergic nuclei (locus coeruleus, ventral tegmental area, median and dorsal raphe) and vegetative hubs (periaqueductal gray and hypothalamus) was uncovered. A more detailed investigation of fatigue has shown an almost identical pattern for mental and physical domains of fatigue (Supplementary Fig. 7). To account for the two recruitment periods of patients with Post-COVID-Condition (PCC), we furthermore added the period as third nuisance covariate in addition to age and sex into our abovementioned models. However, effect-patterns were consistent between models for cognition, olfaction, and fatigue (Supplementary Fig. 8).

## Discussion

Here, we report clinical and (micro-) structural MRI data from a prospective cohort of patients diagnosed with Post-COVID-Condition (PCC)[3] and have described them in comparison to healthy controls (HNC) and unimpaired patients who had previously contracted a COVID-19 infection (UPC). Although FreeSurfer-based analysis did not provide any detectable evidence for global and cortical atrophy,

widespread changes in cerebral microstructure were identified between groups. On a gross scale, a significant volume shift from the membrane-enclosed compartment (i.e., V-extra + V-intra with predominant effects on V-extra) into the microstructural free-water compartment (V-CSF) occurred in the gray matter of patients with PCC and correlated with initial disease severity. However, voxel-based comparisons of V-extra between groups revealed an even more distinct view on the COVID-19-related effect: Whereas a marked reduction in V-extra occurred in the neocortical gray matter and thalamus, increased V-extra was present within the corpus callosum, internal capsule, cerebellum, and brainstem. However, the PCC and UPC group could be differentiated by their different emphasis on this opposite pattern: While the reduced cortical V-extra prevailed in patients with PCC, increased infratentorial V-extra was more pronounced in participants with UPC. To determine microstructural correlates of PCC-associated symptoms after COVID-19 infection, voxel-based associations of V-extra with clinical scores revealed significant correlations between distinct networks and impaired cognitive- or olfactory-performance and fatigue.

The dMRI standard model on which DMI is based was originally developed for white matter, as it implies no exchange between the one-dimensional compartment (V-intra) and the extra-axonal (V-extra) and CSF space[9,10]. While this assumption is validated by the presence of myelin sheaths in white matter, it might not be applicable to gray matter. In fact, recent studies have suggested a non-negligible exchange between dendrites and extracellular space on a scale of ~10 ms[19], 20–60 ms[20], or even <10 ms[21]. Such minute values do not allow a clear distinction between V-intra (neurites) and V-extra (cells and extracellular matrix) in gray matter. This also explains why the intra-neurite volume fraction we obtained in gray matter was rather low (compared to the fraction of neuropil present in gray matter). Therefore, the present finding more likely represents a shift in volume from the membrane-enclosed compartment (V-intra + V-extra) towards V-CSF.

Due to the lack of neuropathological studies on patients with PCC, the mechanistic basis for the observed microstructural changes in the brain remains speculative. In the present study, patients with PCC were characterized by shift in volume from the membrane-enclosed compartment of gray matter (i.e., the compartment of cell bodies, neurites, and extracellular matrix) to the V-CSF (representing interstitial free fluid and perivascular spaces), which could be explained by shrinkage due to degeneration or cell loss[22]. Such changes have been observed not only during healthy aging[23,24], but also in neurodegenerative disorders like Parkinson's disease[25] or amyotrophic lateral sclerosis[26]. However, in a functional brain imaging study using [18]F-FDG PET, no alterations in cerebral glucose metabolism were found in a cohort of patients with PCC that partially overlapped with those in the present study (six patients that underwent PET-imaging were also included in this study) in comparison to controls[5]. As [18]F-FDG PET can detect neurodegenerative diseases even in prodromal states[27,28], the normal molecular imaging findings observed in the aforementioned study strongly argue against significant cellular loss. Hence, cellular shrinkage in the context of normal aging processes would be a valid explanation for our results. In line with this hypothesis, post-mortem transcriptomic analysis of frontal cortex tissue of patients that died from COVID-19 revealed a molecular signature of aging when compared to controls[29]. Although a volume shift from V-extra into V-CSF was also observed in the cortical gray matter of UPC patients, the opposite process in which a volume shift to the cellular and extracellular matrix compartments – particularly in the brainstem and cerebellum – actually prevailed. This increase in V-extra could be explained by the growth of the cellular compartment, due to common neuro-inflammatory processes such as astrocyte swelling or microglia invasion[30,31]. Indeed, evidence for the activation of astrocytes and microglia (including the formation of microglia nodules) has been found in post-mortem examination of COVID-19 patients, particularly in the brainstem and cerebellum[32,33]. Although an increase in V-extra within the brainstem, cerebellum, and corpus callosum was also present in patients with PCC, it was much more attenuated than that observed in the UPC group. This, however, does not necessarily mean that potential inflammatory processes are less relevant for this population. Axonal damage or cellular shrinkage/loss can occur as a consequence of such an inflammatory process and hence further reduce V-extra values[30], causing an intermediate net effect. Based on these considerations, COVID-19 may have elicited a long-lasting inflammatory response that is still detectable in the UPC group, whereas a degenerative (e.g., accelerated aging processes) residual effect prevails in patients with PCC. Again, this hypothesis requires neuropathological verification. Interestingly, the time span between positive SARS-CoV-2 PCR and the cerebral MRI did not explain alterations of gray matter DMI parameters or spatial distribution of V-extra changes between the PCC and UPC groups. Thus, one could speculate on a slow or even non-reversibility of microstructural changes observed here. In line with this assumed chronicity, 85% of patients reporting complaints two months after COVID-19 still reported symptoms one year after their symptom onset[34]. However, longitudinal studies are required to make definite statements on reversibility of microstructural alterations after COVID-19 in general and in patients with PCC.

With respect to functional anatomy, the composition of the affected networks correlated well with the associated clinical scores quantifying PCC-related symptoms. Impaired cognitive performance showed a significant correlation with reduced V-extra within the thalamus and mesiotemporal regions (including the hippocampus and parahippocampal gyrus), particularly in the left hemisphere. The role of the mesiotemporal lobe in memory formation is well defined[35] and in a neurodegenerative disease cohort, hippocampal volumes were shown to have a strong association with MoCA-performance[36]. On the other hand, remote degeneration of thalamic nuclei after left hemispheric stroke was associated with impaired MoCA-performance[37] and lesion models indicate a role for anterograde memory and visuospatial processing[38]. The proposed roles for these structures therefore support our observations that the MoCA-domains memory and visuoconstruction were particularly affected in patients with PCC. With respect to impaired olfaction, a significant correlation was present with reduced V-extra within the orbitofrontal cortical areas, mesiotemporal structures (including inferior-temporal, entorhinal, and parahippocampal cortex, amygdala, and hippocampus), thalamus and cerebellum. Whereas orbitofrontal areas contain the secondary and tertiary olfactory cortex[39], the aforementioned temporal regions belong to limbic structures that are involved in the olfactory modulation of emotional states[40]. The thalamus, on the other hand, receives input from primary olfactory areas and is tightly coupled with the orbitofrontal cortex[41]. Even though the cerebellum does not directly participate in olfactory perception, olfactory-related responses have been recorded using functional MRI, especially those evoked by unpleasant odors[42]. However, it is not possible to determine whether V-extra changes are the cause of impaired olfactory performance, or just the consequence of disrupted peripheral sensory input. SARS-CoV-2 RNA and neuro-inflammation have been detected within the olfactory bulb of patients that died from COVID-19[43], and reduced olfactory bulb volumes were detected in patients with olfactory dysfunction six weeks after COVID-19 infection[44]. Unfortunately we were not able to assess the microstructure of the olfactory bulb or tract, because susceptibility artifacts close to the base of the skull combined with the small volumes of these structures prevented valid measurement of DMI parameters. With respect to fatigue, regions in which reduced V-extra significantly correlated were found to encompass a network which included aminergic brainstem nuclei, autonomic hubs, thalamus, left temporo-occipital structures, right putamen, and cerebellum. Although a thorough review of the structural basis and

pathophysiology of fatigue is beyond the scope of this study, there was good agreement with the literature in terms of the structures typically affected during this state: Dysfunction of monoaminergic signaling related to arousal, sleep/wake or reward systems is commonly discussed as one neurobiological basis of fatigue[45]. Furthermore, the involvement of the right putamen and thalamus would fit with the finding that dysfunction in cortico-striato-thalamo-cortical loops is associated with fatigue in patients suffering from multiple sclerosis[46]. Reduced functional connectivity within the left temporo-occipital cortex (including lingual and fusiform cortex) have been observed in post-stroke fatigue[47] and remembered fatigue[48]. Affection of vegetative hubs such as the hypothalamus and the hypothalamo-pituitary-adrenal axis is also frequently suggested to play a central role in the pathogenesis of fatigue[49]. With respect to the periaqueductal gray, increased activation has been measured after provocation of exercise in patients suffering from chronic fatigue syndrome[50]. In a cohort covering the late subacute phase (on average three months) after COVID-19 infection, an association between fatigue scores and lower fiber density within a widespread network of association, projection, and commissural tracts was detected[51]. However, the comparability to our study is limited by the early time point and lack of information about whether patients fulfilled the WHO criteria for PCC.

Douaoud and colleagues recently described increased diffusion indices within the limbic regions (e.g., anterior cingulate, hippocampal, parahippocampal and orbito-frontal cortex) and striatum of patients four to five months after COVID-19 infection in a large longitudinal MRI-study of the UK Biobank[6]. More precisely, an increment of the diffusion tensor imaging (DTI)-parameter mean diffusivity (MD) and the NODDI-parameter isotropic volume fraction (ISOVF) both point to an increment of free water within the aforementioned regions. In good agreement with these data, we observed in cortical and subcortical gray matter a widespread shift in volume from the extra-neurite (V-extra) into the free-water compartment (V-CSF) in patients after COVID-19 infection (pooled PCC and UPC group; Fig. 2A), nine months after a positive SARS-CoV-2 PCR result. Moreover, Douaoud and colleagues[6] reported a decrease in brain volume and cortical thickness (bilaterally: parahippocampal gyrus, anterior cingulate cortex, and temporal pole; left-sided: orbitofrontal cortex, insula, and supramarginal gyrus) that was associated with cognitive decline. A similar pattern of cortical atrophy was described in a cohort of patients with PCC 11 months after infection, which was also associated with impaired attention, working memory, and processing speed[7]. Furthermore, a reduction in axial and mean diffusivity was detected in supra- and infratentorial white matter, while resting-state fMRI analysis revealed hypoconnectivity between left and right parahippocampal areas as well as between bilateral orbitofrontal and cerebellar areas. Accordingly, it is important to keep in mind that only approximately 10% to 25% of COVID-19 patients develop a PCC[1,2]. As a consequence, and despite the lack of clinical information, it can be assumed that the vast majority of the UK Biobank cohort was clinically asymptomatic and it is unlikely that results were driven by a minority with PCC. Given the similarity of results in terms of cortical atrophy and associated cognitive deficits in patients with PCC[7], it still needs to be determined whether these findings are representative of PCC-associated pathology or are simply a PCC-independent consequence of COVID-19 infection. These considerations further highlight the importance of having a control group that comprises patients who had previously contracted COVID-19 but did not go on to develop a PCC. Moreover, in contrast to the aforementioned studies[6,7], we could not identify cortical atrophy or a reduction in global gray matter volume in our PCC cohort. In contrast to the PCC-cohort reported by Díez-Cirarda and colleagues[7], our sample included fewer patients with a severe course of initial disease. However, given that atrophy was associated with COVID-19

severity[6,7], this could explain the lack of macrostructural changes in our study. Although most participants in the UK Biobank study had a mild course of disease[6], the large number of participants combined with the longitudinal design of the study may have helped to identify a level of atrophy that was under the detection limit of our study.

In addition to its cross-sectional design, the following limitations have to be considered for the interpretation of the present findings. Regarding the microstructure imaging approach, it should be taken into account that the range of V-CSF values (-0.15) where we found changes is slightly more susceptible to dMRI-related noise than other ranges, where the fractions of V-CSF and V-intra + V-extra are more balanced[9,52]. Furthermore, the sampling of the diffusion-weighted sequence with anisotropic voxels with a resolution of 3 mm in the z-direction might introduce a potential bias. However, due to the rather global nature of the present finding, this is unlikely to have substantially hampered the results. Although the HNC group was enrolled without history of a past COVID-19 infection, we cannot rule out with certainty that some participants had previously contracted an infection, as antigen status was lacking. The soundness of our results could have been further strengthened by considering key variables such as socioeconomic status or risk factors of cerebral microstructural integrity (e.g., smoking status), which unfortunately were not available in all cohorts. Although a significantly higher proportion of participants in the PCC cohort suffered from migraine or had arterial hypertension and obesity as risk factors for impaired brain integrity, it is unlikely that the patterns we observed were due to these factors, as these pathological changes impact the brain differently to what we observed[53–55]. It must also be noted that a gap of 7 months divided the recruitment of PCC patients into two periods. However, the period had no influence on the changes in gray matter diffusion MRI parameters (Supplementary Figs. 1 and 2), the spatial distribution and direction of V-extra changes (Supplementary Figs. 5 and 6), as well as the symptom-specific reduction of V-extra (Supplementary Fig. 8). Finally, an additional control group of patients that had contracted influenza or other respiratory tract infections would be required to see if microstructural alterations to gray matter tissue are COVID-19-specific.

In summary, we applied the advanced imaging technique DMI to uncover microstructural changes after COVID-19 infection, with the observation of different patterns in a cohort of patients with and without PCC. A correlation between functional status and imaging data was identified, whereby the presence of PCC symptoms was associated with the affection of specific cerebral networks, suggesting a pathophysiological basis for this syndrome.

## Methods

### Study participants and clinical outcomes

This study complies with all relevant ethical regulations and was approved by the Ethics Committee of the University of Freiburg (EK 211/20). All participants provided written informed consent in accordance with the Declaration of Helsinki and its amendments. The study was registered to the 'Deutsches Register Klinischer Studien (DRKS)' (DRKS00021439). We present exploratory data from a monocentric, prospective cohort of 89 patients (age: 49 [23]; sex: 34/55 males/females - self-reported), who were admitted to the outpatient clinic of the Department of Neurology and Clinical Neuroscience of the University Hospital Freiburg due to neurocognitive symptoms in the chronic phase of COVID-19 infection. Inclusion criteria were: (1) a SARS-CoV-2 infection confirmed by reverse transcription polymerase chain reaction (rt-PCR); (2) fulfillment of diagnostic criteria for Post-COVID-Condition according to WHO criteria (e.g., >3 months after onset of acute COVID-19 infection; symptoms lasting for at least 2 months; relevant impact on everyday functioning)[3]; (3) execution of a cranial MRI. Exclusion criteria were any pre-existing

neurodegenerative disorder and an age below 18 years. One female patient (44 years old) had to be excluded due to MRI artifacts that would have interfered with further data processing (these data were also not included in the demographic analysis). The patients with PCC were recruited in two periods ($n = 62$ between June 2020 and January 2022, period 1; $n = 27$ between August 2022 and October 2022, period 2). For the present analyses, both groups were combined. Both groups were treated with an identical data collection method as mentioned above. Two further groups served as controls: (1) Unimpaired Post-COVID (UPC): a collective of 38 participants (age: 42 [24]; 13/25 males/females – self-reported) in the chronic phase following PCR-confirmed COVID-19 infection, without persistent subjective complaints and enrolled between August and October 2022. The same exclusion criteria were applied (i.e., any pre-existing neurodegenerative disorders, age <18 years and artifacts in imaging data), and the examination and measurement methods were identical to the PCC-group. Although the UPC group was recruited with a delay, it took place in the same time period in which patients with PCC were also included and examined ($n = 27$). (2) Healthy non-COVID (HNC group): an in-house collective of 46 healthy participants (age: 44 [31]; range: 21 to 80 years; 23/23 males/females – self-reported), with no history of COVID-19 infection (obtained from medical records and self-reports) and no significant difference in age to participants in the PCC and UPC groups (Mann–Whitney-U, $P = 0.219$). There were no statistically significant differences in the sex of HNC participants compared to those in the PCC and UPC group ($X^2$, $P = 0.281$; for more detailed information see Table 1). HNC participants were enrolled between June 2020 and January 2022. Patients were examined and surveyed by board certified (AD, JH) or experienced (>6 years of training, NS) neurologists. The study participants did not receive any compensation. The degree of current disability was graded as follows: 0, no relevant restrictions; 1, relevant restrictions but able to work; 2, reduction of workload required; 3, inability to work and/or restriction of daily life activities. Disease severity during the acute stage was assessed according to a modified version of the German definitions[56]: 1, no signs of pneumonia; 2, pneumonia, outpatient treatment; 3, pneumonia, in-patient treatment; 4, acute respiratory distress syndrome (ARDS), endotracheal ventilation in the intensive care unit (ICU). Disease severity was considered to be mild in the case of ambulatory treatment (i.e., 1–2) and severe in patients that required hospitalization (i.e., 3–4). Cognitive functions were assessed with the German version of the Montreal Cognitive Assessment (MoCA version 7.1, www.mocatest.org)[12]. The highest possible global MoCA score is 30, with higher scores indicating better performance. The cut-off score for cognitive impairment was defined as <26[12]. Correction for years of education (YoE) was performed (+ 1 point in case of ≤12 YoE). MoCA domain scores were calculated as the mean of single item scores and comprised subscores for orientation (spatial and temporal orientation), attention (digit span, letter A tapping and subtraction), executive function (trail making, abstraction, and word fluency), visuoconstructive function (cube copying and clock drawing), language (naming, sentence repetition), and memory (delayed word recall). Domain scores were not adjusted for YoE. Fatigue was evaluated using the Würzburg Fatigue Inventory in Multiple Sclerosis (WEIMuS)[57], a self-rating questionnaire for symptoms of physical and cognitive fatigue. In addition, the presence of depression was assessed using the Geriatric Depression Scale-15 (GDS)[58] and olfaction was assessed using Burghart-Sniffin'-Sticks® (Burghart Messtechnik GmbH, Wedel, Germany)[59]. Ammonium was used to assess trigeminal function (normosmia: 11–12 correctly identified odors; hyposmia: 7–10 correct odors; anosmia: ≤6 correct odors).

## Cerebral MRI

**MRI acquisition.** MRI was performed with a 3 Tesla scanner (MAGNETOM Prisma, Siemens Healthcare, Erlangen, Germany) with a 64-channel head and neck coil. T1-weighted (T1w) images were acquired with a three-dimensional (3D) magnetization-prepared 180° radio-frequency pulses and a rapid gradient-echo (MP-RAGE) sequence (repetition time: 2500 ms, echo time: 2.82 ms, flip angle: 7°, TI = 1100 ms, GRAPPA factor = 2, 1.0 mm³ isotropic voxels, 192 contiguous sagittal slices). The diffusion weighted sequence was acquired with the following parameters: axial orientation, 42 slices, voxel size $1.5 \times 1.5 \times 3$ mm³, TR 2800 ms, TE 88 ms, bandwidth 1778 Hz, flip angle 90°, simultaneous multi-band acceleration factor 2, GRAPPA factor 2, 58 diffusion-encoding gradient directions per shell with b-factors 1000 and 2000 s/mm², and 15 non-diffusion weighted images (interleaved during diffusion-encoding directions); this resulted in a total of 131 images.

**Calculation of DMI parameters.** Data processing was implemented within our in-house post-processing platform NORA: Pre-processing of diffusion-weighted images included a denoising step[60], followed by correction of the Gibbs-ringing artifacts[61] and up-sampling to isotropic resolution of 1.5 mm³. Microstructural diffusion metrics were estimated using a Bayesian approach that determines the three components of a white matter-based tissue standard model[8–10,62–64]: 1. The free water/CSF fraction (V-CSF) in which molecules randomly move at the distance of their diffusion length (in the range of tenths of a micrometer). 2. The volume fraction within neuronal processes (i.e., axons and dendrites; V-intra), with almost one-dimensional molecule diffusion due to tight membrane borders. 3. The volume fraction outside of axons or dendrites (V-extra), characterized by an intermediate constraint to molecule diffusion representing the cellular compartment and the extracellular matrix. Although primarily developed for white matter, multi-compartment diffusion MRI (dMRI) metrics have also been successfully employed to investigate gray matter[22,65–67]. T1w imaging datasets were automatically segmented into white matter, gray matter, and cerebrospinal fluid (CSF) using CAT12 (http://www.neuro.uni-jena.de/cat/), and dMRI images were coregistered to the T1w images. Validity of co-registration between dMRI images and T1w-derived tissue probability values (TPV) were manually confirmed. Further quality control was performed by visually inspecting each individual DMI map and CAT12 segmentation. For the extraction of DMI-parameters of whole brain, gray and white matter compartments were defined based on the respective tissue probability value map thresholded >0.6, as provided by the CAT12 segmentation.

**Spatial normalization and voxel/region-wise comparisons.** To further investigate by voxel-wise analysis the spatial distribution of PCC-related changes on both structural and dMRI parameters, images were spatially normalized by CAT12 and the diffeomorphic anatomical registration using the exponentiated lie algebra (DARTEL) method[68]. The diffeomorphic warp was used to transfer the quantitative dMRI maps to the Montreal Neurological Institute (MNI) space. Images were smoothed with a 3 mm full-width at half-maximum (FWHM) Gaussian kernel. For controls (HNC- and UPC-group), MRIs were acquired over the same period of time, with the same scanner, same coil, and same sequence parameter settings. As implemented in the Statistical Parametric Mapping-Voxel-Based Morphometry (SPM-VBM) 8-Toolbox, voxel-based group comparison of whole-brain DMI parameters was performed using a parametric multiple regression model and threshold-free cluster enhancement[69] (https://github.com/markallenthornton/MatlabTFCE), whereas the parameters age and sex served as nuisance covariates. The family-wise error (FWE)-method was employed to correct for multiple comparisons (i.e., across voxels). Cortical morphometry including cortical thickness, cortical surface area, and gray-matter volume was further compared between groups using the FreeSurfer V6.0 toolbox with default

parameters[14]; cortical parcellation was performed thereafter using the Desikan-Killiany atlas (DKT)[16]. Age and sex served as nuisance covariates and the FWE method was applied to correct for multiple comparisons (i.e., across DKT-ROIs). The total gray-matter volume was defined as the sum of the individual gray-matter volumes of all cortical ROIs. To correlate clinical outcomes with DMI parameters in patients with a previous COVID-19 infection (i.e., PCC and UPC), voxel-based analyses were performed, where V-extra was used as a dependent variable, and threshold-free cluster enhancement was applied. Age and sex served as nuisance covariates and FWE-correction was applied to control for multiple comparisons (i.e., across voxels). The resulting areas of significance were converted to a binary mask and anatomical allocation was performed secondarily using various neuroanatomical atlases as references[15–18].

### Statistical analysis

Statistical analyses were performed using R (version 4.1.2, https://www.R-project.org/) and SPSS, Version 25 (IBM, Ehningen, Germany). No statistical method was used to predetermine sample size. The Shapiro-Wilk test was used to assess the distribution of data. In the case of normal distribution, data were presented as mean values (standard deviation) and $t$ tests were used for group comparison. Non-normally distributed data were presented as the median value [inter quartile range] and the non-parametric two-tailed Mann–Withney-U- or Kruskal–Wallis-tests were applied. For comparison of comorbidities, Fisher's Exact test was used. Equal variance was formally tested by Levene's Test. Correlations between clinical data (i.e., current disability, disease severity, MoCA-performance, WEIMuS, olfactory performance, and GDS-15) were identified using Spearman's rank correlation test, where Bonferroni-correction has been applied across 15 comparisons. For comparisons between groups, two-tailed ANCOVAs were performed with the nuisance covariates age and sex, and Tukey's honestly significant difference served as the post-hoc test. For ANCOVAs comparing whole-brain gray and white matter DMI parameters, Bonferroni-correction was applied to account for multiple comparisons as follows: For whole-brain gray and white matter DMI parameters, correction has been applied across 6 comparisons (3 [V-extra, V-intra, V-CSF] x 2 [white and gray matter]). In the whole-brain DMI-analysis that further accounted for disease severity or delay, correction has been applied across 12 comparisons (3 [V-extra, V-intra, V-CSF] x 2 [white and gray matter] x 2 [effect of group and disease severity]). Furthermore, we performed a complete case analysis for all models and examined residuals for internal validation.

### Reporting summary

Further information on research design is available in the Nature Portfolio Reporting Summary linked to this article.

### Data availability

The anonymized data generated in this study have been deposited in the Dyrad database (https://doi.org/10.5061/dryad.kkwh70s9g)[70]. As we did not obtain consent to publish information that identifies individuals, we aggregated age into 5-year categories. The raw MRI data may contain information that could compromise the participants' privacy and can only be made available on request from the corresponding author (JAH, jonas.hosp@uniklinik-freiburg.de; response within four weeks; data might not be used to identify individual participants). The employed cerebral atlases are available via LeadDBS (V2.5; https://www.lead-dbs.org/download/).

### Code availability

The code used in this study is publicly available via https://bitbucket.org/reisert/baydiff/wiki/Home for the DMI processing, https://github.com/spisakt/pTFCE for TFCE with FWE-correction, https://surfer.nmr.mgh.harvard.edu/fswiki/DownloadAndInstall for FreeSurfer V6.0.

https://www.fil.ion.ucl.ac.uk/spm/software/download/ for SPM, and http://www.neuro.uni-jena.de/cat for CAT12. The statistical analysis code used in our study has been deposited in the Zenodo database (https://doi.org/10.5281/zenodo.8288991)[71].

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

## Acknowledgements

A.R. and N.S. are supported by the Berta-Ottenstein-Programme for Clinician Scientists, Faculty of Medicine, University of Freiburg. J.A.H. and A.D. are supported by the Berta-Ottenstein-Programme for Advanced Clinician Scientists, Faculty of Medicine, University of Freiburg.

## Author contributions

Author contributions J.A.H., M.R., A.D., V.G., E.K., H.M., S.A., C.F.W., D.W., S.R., H.U., C.W., N.S., and A.R. created, extracted, and organized the imaging and clinical data. J.A.H., M.R., N.S., and A.R. carried out the imaging analyses. J.A.H., M.R., and A.R. interpreted the results. J.A.H. and A.R. wrote the paper. All co-authors revised the manuscript.

## Funding

## Competing interests

H.U. is co-editor of Clinical Neuroradiology, member of the Advisory Board of Biogen and received honoraria for lectures from Biogen, Eisai, and mbits. E.K. is shareholder of and received fees from VEObrain GmbH, Freiburg, Germany. N.S. received honoraria for lectures sponsored by Abbvie and Novartis. All other authors report no competing interests.
