## [Peer Review File · Nature Communications]

Cerebral microstructural alterations in Post-COVID-condition are related to cognitive impairment, olfactory dysfunction and fatigueReviewers' comments:

Reviewer #1 (Remarks to the Author):

In this study, Hosp, Reisert and colleagues have assessed structural and microstructural brain abnormalities in 62 long covid patients according to the WHO definition, as well as brain correlates for olfaction, disability, cognitive performance, disease severity, fatigue and depression. This is a timely study, adding to the objective evidence accumulating for long covid, with a well-characterized patient population. As it stands however, there are some methodological considerations that cast doubt over the main results presented, and this study lacks key methodological - i.e., participants recruitment, imaging processing, statistics - details and justifications.

1. Control group. While the long covid patient population is well-characterized clinically, precious little information is given about the control group. There are only two demographic variables reported (age and sex). MoCA is mentioned as being "unimpaired", but no quantification given. Worse of all, nothing is said either about the controls' covid diagnoses (or absence thereof) and how they are indeed controls for this long covid group, or about the type of imaging they've received and on which scanner (imaging parameters/scanning information is only given for the patients).

Altogether, this perhaps also suggests something more aggravating: that this control group, defined in the text as "an in-house control collective of 47 healthy subjects", was not part of the same cohort, and indeed, was acquired at a completely different time, maybe even before the beginning of the pandemic, which would explain why there is no need to check for possible coronavirus infection in this control group. If that is the case, then all group comparisons (as opposed to correlations within the patient group only) are unfortunately void and uninterpretable. The main contrast of interest, comparing controls and patients, would be entirely collinear with the necessary nuisance covariate of cohort/time/scanner/acquisition. With an entire control group not acquired as part of the same cohort and same imaging protocol, one simply cannot conclude that the findings are in any way specific to long covid, rather than e.g., scanner/acquisition differences, scanner drifts, effects of isolation/anxiety/lockdown, etc.

If this is not the case however, and the control group was actually part of the same imaging study, acquired over the same period of time, with the same scanner and same imaging protocol, then it is key for the authors to report this clearly, as well as to expand on the quantitative information for this control group (how do they compare to the covid group cognitively, how do they compare for well-known covid risk factors, SES, etc.).

2. Clinical correlates. The title of this study is somewhat misleading. It is certainly not possible with the data and statistical analyses provided in this study to establish whether "altered grey matter" is either causative, or an epiphenomenon in (long) covid. The absence of correlation of one single imaging variable with "clinical measures" does not constitute a proof that there is none. In fact, this too is somewhat misleading: there is a clear correlation between the diffusion measure the authors focus on post-hoc: Vextra - and a comparable NODDI measure, Vic, too - with disease severity. It is unclear why the disease severity score is considered as part of the "clinical readouts" (Table 2) and the "clinical outcomes" (Methods - p 26), but, when it comes to this significant correlation, it is no longer considered a clinical measure. It seems also that this clinical variable is only available in 18 out of the 62 patients (which could perhaps be labelled more clearly than as a footnote in Table 2), but it is not made clear why that might be.

Concerning the statistical model(s) used to assess correlations between clinical variables and Vextra, the authors do not justify why the disease severity score becomes a nuisance covariate, instead of a variable of interest. It is also completely unclear how this score is included in the model to assess the effects of olfaction/depression/etc., considering that it is only available in 18 out of 62 of the patients. It is also not explained why the authors choose to add, as an additional nuisance covariate, the time

between the positive PCR test and MRI scan, i.e., how long before the MRI the patients were infected. It might be a completely legitimate model, but the authors should explain the kind of questions they are asking by adding one (or two) of these covariates in their model(s) and consequently in which light they can interpret the answers they get - in this case, a lack of significant correlation with olfaction, depression, fatigue and cognitive performance, above and beyond disease severity and regardless of the time that has passed since contracting covid. It would also be of interest to assess the effect of that period of time between PCR and MRI itself.

3. Imaging processing and statistical analyses. It is at the moment not always completely clear, both in the Methods and Results, when group comparison and correlation analyses have been made voxel-wise, over a global mask of brain/grey/white matter or based on regions-of-interest, and if so, which regions and from which atlas. For instance, the Desikan-Killiany cortical atlas is mentioned, but also the AAL3 atlas. The latter can be problematic, as it is based on the functional data of a single subject. Probabilistic atlases, or individual identification (such as using the Desikan-Killiany) are maybe preferable. For the creation of global masks, the steps for their creations should also be detailed.

Details for the Freesurfer and the SPM-VBM pipelines are also lacking and this section should be expanded. In addition, it is not known whether the same amount of smoothing is used for both structural and diffusion images. Could the authors also please justify the use of a small Gaussian kernel of 3 mm (esp. when their initial diffusion MRI data is of 3 mm in thickness before upsampling, which should perhaps call for a larger kernel of at least 6 mm)?

The authors also mention applying Bonferroni corrections, but as far as I can see, it is never specified across what, and how many comparisons are then corrected for (how many voxels, parcels/regions, number of clinical variables, etc.).

The definition and numbers of variables in the models vary, and as mentioned before, without full justification. In different parts of the text for seemingly the same analyses, the same variables are described in turn as "nuisance covariates" or "covariates" (of interest). Presumably the same linear models were used, and it only depends on whether the authors focused their attentions to the betas/stats associated with one variable or another, but some consistency would be welcome.

Finally, age x group interactions are not taken into account at the moment, when it would appear from previous publications that there is a differential effect of age associated with covid. The authors should consider adding this interaction in their model.

4. Direction of effects. Different parts of the text confusingly refer to presumably the same findings in opposite way. For instance, Figure 2B refers to a significant *decrease* in Vextra in the covid group: "Voxels with significantly decreased V-extra in Post-COVID-syndrome patients were indicated by blue shading", whereas p9, line 25 of the Results mentions a significant *increase* in Vextra in the covid group: "Areas with significantly increased V-extra in Post-COVID patients encompass widespread cortical (...)".

In line with this, when looking at Figure 2A, there appears to be strong, opposite effects (in blue color vs in red color). Have the authors explored the significance of the effects in both directions? In Figure 2B, only those areas corresponding to the blue effects seem to have been investigated. Even if the opposite effects might be down to artefacts, this needs to be explored and discussed.

5. Data availability. I do not know whether a provision has been made with the Editors, but the current approach, namely "Data and code are available from the authors upon reasonable request and approval of the ethics committee" is regrettably inadequate, particularly after checking the NPG requirements:

A condition of publication in a Nature Portfolio journal is that *authors are required to make materials,

data, code, and associated protocols promptly available to readers without undue qualifications*. Any restrictions on the availability of materials or information must be disclosed to the editors at the time of submission. Any restrictions must also be disclosed in the submitted manuscript.

<https://www.nature.com/nature-portfolio/editorial-policies/reporting-standards>

6. Minor issues. "MRI" is sometimes misspelled "MRT" in " Δ positive PCR - cMRT (days)". "Gust" is obsolete.

Reviewer #2 (Remarks to the Author):

Thank you for asking me to look at this paper which makes some interesting imaging observations in relation to Post-Covid Syndrome. When I agreed to review the paper I did not realise quite how involve the imaging studies were, and this is not my area of expertise. I therefore can find myself to some general observations. The main observation in this study was gray matter volume shifts which were compatible with accelerated ageing and comparable to the previous studies. They found these alterations correlated with severity of initial infection but not with clinical outcomes in terms of olfactory performance, fatigue and cognition. They therefore concluded that they were an epiphenomenon of Covid-19 rather than the structural basis of the longer-term complications. For me there is a flaw in the logic here. Just because A does not directly correlate with B, it does not therefore follow that A cannot have any causal relationship with B. It may be that there are other compounding causal factors involved which have not been measured, and thus not been allowed for. I think they allowed for age and sex, but what about other potential factors like co-morbidities, smoking, alcohol use, etc. I think it is worth at least discussing this even if you are not able to study it and allow for it. Secondly they out in the discussion that it would've been helpful to have a control group of previous Covid 19 patients who do not now have postcode syndrome, but in whom they perform the same imaging studies. I would have thought this was essential to do to support their argument that the structural changes they see are simply an effect of severe Covid and have no relationship with the long-term post Covid problems. I think the fact that they did not include any controls at all it's regrettable it significantly we can see the paper

I am sorry that I am only able to provide this limited report, but detailed imaging studies are really beyond my area of expertise.

Reviewer #3 (Remarks to the Author):

The manuscript presents brain tissue microstructure changes, obtained with diffusion MRI, in gray matter in patients with post-Covid syndrome. The work adds to the on-going debate about the microstructural (cellular-level) origins of the syndrome, and specifically stipulates that the recently found association between "free water" fraction increase and the disease does not imply that the post-Covid syndrome originates due to such changes.

While I find the results intriguing and timely, I have a few suggestions. The major one is about assigning of model's "compartments" to intra- and extra-neurite. The dMRI "Standard Model" (SM) the authors are using has been developed for white matter, as it implies no exchange between the one-dimensional compartment (originally ascribed to axons) and the extra-axonal space. (The third compartment, CSF, corresponds to macroscopic partial-volume inclusions in a voxel.) While SM has been previously applied to gray matter, lumping together axons and dendrites into a "neurite" compartment non-exchanging with extra-cellular space has never been justified. In fact, recent studies have been pointing at a non-negligible exchange between dendrites and extra-cellular space,

on the scale of ~ 10 ms (Williamson eLife 2019), 20-60ms (Jelescu Neuroimage 2022), or even < 10 ms (Olesen Neuroimage 2021). In any case, the exchange time seems to be smaller or of the order of the diffusion time in clinical dMRI. Not surprisingly, the intra-neurite volume fraction the authors obtain is $< 25\%$, much below that of neuropil in gray matter. It may as well be that this fraction corresponds mostly to axons (that are inevitably present in the voxels due to partial-volume effects), although the dendrites may somewhat affect this fraction depending on the exchange rate that, alas, has still not been determined reliably. While I do not think this issue should disqualify the work, the authors should be more careful about interpreting their results in terms of specific compartments, and a discussion about exchange and applicability of SM-like models without exchange is required.

Other issues:

1. Placing the work in context: DMI is mentioned from the very beginning, yet it's referenced somewhere in the middle (refs 12,13), with particular refs being somewhat tangential to this work (ref 12 is on tractography, and ref 13 is about diffusion time-dependent effects beyond the adopted SM picture). I suggest referring to recent review(s) on the subject. Furthermore, tissue microstructure imaging with dMRI relies on many models, whereas the authors adopt a particular one, the SM (defined above). Its brief description, assumptions (ie Gaussian diffusion and absence of exchange), and key SM references, should be part of the main text, as the use of this model underpins all the interpretations.
2. Sensitivity to volume fractions: Please show results of noise propagation with the SNR you have in your data, for the specific protocol, and the parameters (volume fractions) that are highlighted in the work. Can you quantify the spurious correlations for the two independent volume fractions inevitable due to the nonlinear character of the model and the degeneracies in its parameter estimation? This is important since it's known that estimating the free water fraction is particularly difficult in a limited 2-shell protocol.
3. Protocol: it's stated that there are 65 diffusion directions, yet 58 images per shell. Should there be 65 images per shell then?
4. Can you comment about the validity of upsampling 3mm to 1.5mm in a very narrow cortex?

Authors' Response to Reviewers' Comments:

Article Title: Cerebral microstructural alterations in Post-COVID-condition are related to cognitive impairment, olfactory dysfunction and fatigue

General remark: We would like to thank the reviewers for their thorough review and their valuable and thoughtful comments on our manuscript. By means of this important advice, we included an additional cohort of subjects that passed COVID-19 without having any actual complaints ("Unimpaired Post-COVID", UPC-group), expanded the cohort of patients suffering from "Post-COVID-Condition" (PCC-group) and deepened the characterization of the control group of subjects without any history of COVID-19 infection ("Healthy Non-COVID", HNC-group). This however led to new analyses that fundamentally changed the key messages of this study.

In brief, we now show that COVID-19 induced a specific pattern of mutual volume shifts between the extraneurite compartment (V-extra, i.e. somata, extracellular matrix) and the free water fraction (V-CSF, i.e. cerebrospinal fluid, perivascular spaces): whereas a V-extra decrease occurred in neocortical gray matter and thalamus, increased V-extra was present within corpus callosum, internal capsule, cerebellum and brainstem. Moreover, PCC- and UPC-patients differed regarding their emphasis on this pattern. Correlations between V-extra and clinical scores revealed affection of symptom-specific networks associated with impaired cognition, or reduced olfactory-performance and fatigue. Thus, this study now revealed cerebral microstructural changes after COVID-19 with distinguishable patterns in patients with and without PCC. PCC-symptoms were associated with affection of specific networks, providing a potential biomarker of this syndrome.

In the following, we provide a point-by-point answer describing how we addressed the individual comments of the reviewers for the creation of the new manuscript (answers are reported in red, and changes to the manuscript in *italic red*). As the current version had to be largely rewritten, we waived to highlight all applied changes.

Reviewers' comments:

Reviewer #1 (Remarks to the Author):

In this study, Hosp, Reisert and colleagues have assessed structural and microstructural brain abnormalities in 62 long covid patients according to the WHO definition, as well as brain correlates for olfaction, disability, cognitive performance, disease severity, fatigue and depression. This is a timely study, adding to the objective evidence accumulating for long covid, with a well-characterized patient population. As it stands however, there are some methodological considerations that cast doubt over the main results presented, and this study lacks key methodological - i.e., participants recruitment, imaging processing, statistics - details and justifications.

We thank the reviewer for the evaluation of our work. We addressed the raised concerns as stated below and are confident that doing so, we overcame the methodological considerations.

1. **Control group.** While the long covid patient population is well-characterized clinically, precious little information is given about the control group. There are only two demographic variables reported (age and sex). MoCA is mentioned as being “unimpaired”, but no quantification given. Worse of all, nothing is said either about the controls’ covid diagnoses (or absence thereof) and how they are indeed controls for this long covid group, or about the type of imaging they’ve received and on which scanner (imaging parameters/scanning information is only given for the patients). Altogether, this perhaps also suggests something more aggravating: that this control group, defined in the text as “an in-house control collective of 47 healthy subjects”, was not part of the same cohort, and indeed, was acquired at a completely different time, maybe even before the beginning of the pandemic, which would explain why there is no need to check for possible coronavirus infection in this control group. If that is the case, then all group comparisons (as opposed to correlations within the patient group only) are unfortunately void and uninterpretable. The main contrast of interest, comparing controls and patients, would be entirely collinear with the necessary nuisance covariate of cohort/time/scanner/acquisition. With an entire control group not acquired as part of the same cohort and same imaging protocol, one simply cannot conclude that the findings are in any way specific to long covid, rather than e.g., scanner/acquisition differences, scanner drifts, effects of isolation/anxiety/lockdown, etc. If this is not the case however, and the control group was actually part of the same imaging study, acquired over the same period of time, with the same scanner and same imaging protocol, then it is key for the authors to report this clearly, as well as to expand on the quantitative information for this control group (how do they compare to the covid group cognitively, how do they compare for well-known covid risk factors, SES, etc.).

We agree with the reviewer that description of the control group within the initial manuscript was misleading and kept too short. Healthy patients without a history of passed COVID-19 infection (i.e. the “Healthy Non-COVID” or “HNC”-group of the revised manuscript; n = 47) were part of the same imaging study, acquired over the same period of time, with the same scanner, same imaging protocol and same

coil. As suggested by the reviewer, we added detailed information on comorbidities and MoCA performance (i.e. median (IQR): 28.7 (1.5); range 26 to 30) into the revised manuscript.

Moreover, guided by the concerns of the reviewers 1 and 2, we expanded the population of patients with "Post-COVID-Condition" ("PCC"-group, from 62 to 89 individuals) and included a second control group consisting of 38 subjects that passed COVID-19 without having longer-lasting complaints (i.e. "Unimpaired Post-COVID" or "UPC"-group. Again, UPC-subjects were part of the same imaging study, acquired over the same period of time, with the same scanner, same imaging protocol, and same coils. Regarding age and sex, there were no significant differences compared to HNC- and PCC-groups. Furthermore, clinical examination of UPC-subjects was performed in the same manner as for PCC-patients.

These fundamental changes in the revised version of our manuscript enabled new analyses allowing fundamentally novel insights into COVID-19 related changes on brain's microstructure. COVID-19 induced a specific pattern of mutual volume shifts between the extraneurite compartment (V-extra, i.e. somata, extracellular matrix) and the free water fraction (V-CSF, i.e. cerebrospinal fluid, perivascular spaces): whereas V-extra decrease occurred in neocortical gray matter and thalamus, increased V-extra was present within corpus callosum, internal capsule, cerebellum and brainstem. Moreover, PCC- and UPC-patients differed regarding their emphasis on this pattern. Correlations between V-extra and clinical scores revealed affection of symptom-specific networks associated with impaired cognition, or reduced olfactory-performance and fatigue.

In summary, this study now revealed cerebral microstructural changes after COVID-19 with distinguishable patterns in patients with and without PCC. PCC-symptoms were associated with affection of specific networks, providing a biomarker of this syndrome.

In the following, we provide a selection of changes within the revised manuscript how we addressed the reviewers concerns:

A detailed description of the UPC-group and a comparison with the PCC-group was provided in the **Tables 1** and **2**. A detailed description of the HNC-group (demographic data, comorbidities and MoCA-performance) was furthermore provided within the **Supplementary Table 1**.

Methods section page 31: *"Two further groups served as controls: 1) "Unimpaired post-COVID" (UPC): a collective of 38 subjects (age: 42 [24]; 13/25 males/females) in the chronic phase after PCR-confirmed COVID-19 infection without persistent subjective complaints. 2) "Healthy non-COVID" (HNC-group): an in-house control collective of 46 healthy subjects (age: 41 [32]; range: 21 to 80 years; 23/23 males/females) with no history of passed COVID-19 that did not significantly differ in age to PCC- and UPC-groups (Mann-Whitney-U, $P = 0.21$). Regarding sex, there was also no significant difference compared to PCC- (χ^2 , $P = 0.82$) and UPC-group (χ^2 , $P = 0.28$; for more detailed information see **Supplementary Table 1**)."*

Methods section page 33: *“For controls (HNC- and UPC-group), MRIs were acquired over the same period of time, with the same scanner, same coil and same sequence parameter settings.”*

Result section page 7: *“Furthermore, 38 subjects (age 42 [24] years; range: 25 to 62 years; 25 females) with passed COVID-19 infection but without lasting subjective impairment (i.e. “Unimpaired Post-COVID”-group; UPC) were enrolled. Detailed demographic and clinical information is provided in **Tables 1 and 2**. Compared to the PCC-group, no significant differences were present for age, sex, and the delay between positive PCR and imaging (all $P > .05$). Interestingly, a severe course of acute COVID-19 was more frequent in the PCC- compared to the UPC-group (χ^2 , $P < 0.001$). Regarding clinical readouts, UPC-subjects performed significantly better in MoCA (Mann-Whitney-U, $P = 0.003$) and olfactory testing (Mann-Whitney-U, $P < 0.001$) and were significantly less affected in GDS-15 and WEIMuS (both Mann-Whitney-U, $P < 0.001$).”*

2. Clinical correlates. The title of this study is somewhat misleading. It is certainly not possible with the data and statistical analyses provided in this study to establish whether “altered grey matter” is either causative, or an epiphenomenon in (long) covid. The absence of correlation of one single imaging variable with “clinical measures” does not constitute a proof that there is none.

We agree with the concerns raised by the reviewer - indeed the absence of correlation of one variable with another does not constitute proof that there is none. As mentioned above, the novel analyses now indicated that clinical PCC-symptoms were associated with microstructural changes in symptom-specific networks. As this changed the key message of our study, we saw the requirement for a new title of the manuscript, which is now: “Cerebral microstructural alterations in Post-COVID-condition are related to cognitive impairment, olfactory dysfunction and fatigue.”

In fact, this too is somewhat misleading: there is a clear correlation between the diffusion measure the authors focus on post-hoc: Vextra - and a comparable NODDI measure, Vic, too - with disease severity. It is unclear why the disease severity score is considered as part of the “clinical readouts” (Table 2) and the “clinical outcomes” (Methods – p 26), but, when it comes to this significant correlation, it is no longer considered a clinical measure.

The “disease-severity score” (DSS) was considered as a clinical readout, though - on the other hand - it does not constitute a symptom directly associated with the Post-COVID-Condition. Regarding “whole brain” GM- and WM-changes between microstructural compartments (i.e. decrease of V-extra and increase of V-CSF), the DSS was a significant factor in our statistical model. However, group effects remained significant when it was included as a nuisance covariate. Thus, based on whole-brain metrics, a volume shift from V-extra into V-CSF is partially - but not exclusively - driven by the severity of the initial disease. On the other hand, the implementation of the DSS as a nuisance covariate did not affect the spatial distribution of the effects in the voxel-based comparisons (see Supplementary Figure 1). Thus, we conclude that the severity of the initial disease is rather a potential risk factor for the development of a PCC after SARS-CoV-2 infection (in line with a large population-based study;

36229057) than a clinical measure related to PCC. Thus, adding the DSS as a covariate into the models investigating associations between measures of PCC-symptoms (i.e. cognition, olfaction, fatigue or depression) would not add any information with respect to the anatomy of affected networks in PCC. We hope that these considerations justify our choice to discard this measure for correlation with clinical outcome measures of PCC.

Result section page 9: *“To address the impact of severity of initial disease, we included the disease severity score as a nuisance covariate (together with “age” and “sex”) into ANCOVAs comparing whole-brain gray matter parameters between the PCC- and UPC-group. While disease severity contributed significantly to V-extra decrease ($P = 0.004$; $t = -3.48$) and V-CSF increase ($P = 0.078$, $t = 3.29$), “group”-effects still remained significant for both, V-extra ($P = 0.005$; $df: 168$; $t = 3.44$) and V-CSF ($P = 0.048$; $df: 168$; $t = -2.65$). Thus, based on whole-brain metrics, a volume shift from V-extra into V-CSF could be measured for gray matter tissue that is partially - but not exclusively - driven by severity of initial disease.”*

Result section page 10, bottom: *“Although “disease severity” significantly affected whole gray matter V-extra (see above), its implantation into the model as nuisance covariate did not change the spatial distribution of the found results (Supplementary Figure 1).“*

It seems also that this clinical variable is only available in 18 out of the 62 patients (which could perhaps be labelled more clearly than as a footnote in Table 2), but it is not made clear why that might be.

We regret this incorrect statement in Table 2. There, it was indeed stated that the "disease-severity" was only present in 18 patients. In fact, data on disease-severity are available for all PCC patients of the old ($n = 62$) and new ($n = 89$) cohort - and also for the entire novel UPC-group.

3. Concerning the **statistical model(s)** used to assess correlations between clinical variables and Vextra, the authors do not justify why the disease severity score becomes a nuisance covariate, instead of a variable of interest. It is also completely unclear how this score is included in the model to assess the effects of olfaction/depression/etc., considering that it is only available in 18 out of 62 of the patients.

Please see also the previous responses to point 2 related to the disease-severity score (DSS): the parameter was present for the entire cohort and not - as incorrectly stated in Table 2 of the initial manuscript - only for eighteen patients. This parameter was not considered as a target variable because it is not a symptom related to the Post-COVID-Condition. Rather, it is a risk factor for the occurrence of microstructural changes and the development of a PCC - in line with the results of a large population-based study (36229057). Thus, adding the DSS as a covariate into the models investigating associations between measures of PCC-symptoms (i.e. cognition, olfaction, fatigue or depression) would not add any information with respect to the anatomy of affected networks in PCC.

It is also not explained why the authors choose to add, as an additional nuisance covariate, the time between the positive PCR test and MRI scan, i.e., how long before the MRI the patients were infected. It might be a completely legitimate model, but the authors should explain the kind of questions they are asking by adding one (or two) of these covariates in their model(s) and consequently in which light they can interpret the answers they get - in this case, a lack of significant correlation with olfaction, depression, fatigue and cognitive performance, above and beyond disease severity and regardless of the time that has passed since contracting covid. It would also be of interest to assess the effect of that period of time between PCR and MRI itself.

We absolutely agree with the reviewer - as the time between PCR and MRI-acquisition was not different between PCC- and UPC-group, we did not include this variable as a covariate in the revised version of our manuscript.

3. Imaging processing and statistical analyses. It is at the moment not always completely clear, both in the Methods and Results, when group comparison and correlation analyses have been made voxel-wise, over a global mask of brain/grey/white matter or based on regions-of-interest, and if so, which regions and from which atlas. For instance, the Desikan-Killiany cortical atlas is mentioned, but also the AAL3 atlas. The latter can be problematic, as it is based on the functional data of a single subject. Probabilistic atlases, or individual identification (such as using the Desikan-Killiany) are maybe preferable. For the creation of global masks, the steps for their creations should also be detailed.

Preparing the revised version of our work, we kept the reviewer's suggestion in mind and streamlined the methods: 1. Voxel-wise correlations were carried out to investigate group differences in diffusion metrics and associations with clinical symptoms. 2. Anatomical allocations of these group differences and associations were performed with an atlas-based approach. Here the Desikan-Killiany atlas (DKT, PMID 16530430) and the JHU White Matter Parcellation Map III atlas (PMID 19385016) were used. Only for voxels significantly associated with the fatigue score (i.e. WEIMuS), the Human Motor Thalamus atlas of Ilinsky and colleagues (30023427) and the Harvard Ascending Arousal Network atlas of Edlow and colleagues (22592840) were additionally used due to their more detailed parcellation of diencephalic and brainstem structures. 3. The DKT was furthermore used for the investigation of FreeSurfer-derived metrics on cortical thickness, cortical surface area and gray matter volumes. 4. In addition, we added information about the extraction of DMI metrics from whole brain gray and white matter. Here, compartments were defined based on the respective tissue probability value map thresholded >0.6 as provided by the CAT12 segmentation.

Thus, compared to the initial version of the manuscript, different atlases were employed. In the old version, the AAL3 was only used for anatomical correlation of the voxel-wise results as we share the reviewer's concern regarding its single subject evidence.

In the following, we provide a selection of changes within the revised manuscript how we addressed the reviewers concerns:

Results page 10: *“To display the spatial distribution of changes in V-extra across groups, we performed voxel-wise comparisons...”*

Results page 11: *“To search for microstructural correlates of PCC-associated symptoms, we performed voxel-based analyses...”*

Results page 11: *“To obtain a better coverage of brainstem nuclei and diencephalic structures, further atlases were used for anatomical allocations (30023427; 22592840).”*

Methods page 33: *“For the extraction of DMI-parameters of “whole brain” gray and white matter compartments were defined based on the respective tissue probability value map thresholded >0.6 as provided by the CAT12 segmentation.”*

Methods page 33: *“Cortical morphometry including cortical thickness, cortical surface area, and gray matter volume was furthermore compared between groups using the FreeSurfer toolbox after cortical parcellation according to the Desikan-Killiany atlas (DKT)”*

Methods page 34: *“Resulting significant areas were converted to a binary mask and anatomical allocation was performed secondarily using the following atlases (1653043; 19385016; 30023427; 22592840).”*

Details for the FreeSurfer and the SPM-VBM pipelines are also lacking and this section should be expanded. In addition, it is not known whether the same amount of smoothing is used for both structural and diffusion images.

More information on the FreeSurfer and VBM-pipelines were added:

Methods page 33: *“As implemented in the Statistical Parametric Mapping-Voxel-Based Morphometry (SPM-VBM) 8-Toolbox, voxel-based group comparison of the whole-brain DMI parameters was performed using a parametric multiple regression model and threshold-free cluster enhancement 64 (<https://github.com/markallenthornton/MatlabTFCE>) whereas the parameters “age” and “sex” served as nuisance covariates. The Familywise Error (FWE)-method was employed to correct for multiple comparisons. Cortical morphometry including cortical thickness, cortical surface area, and gray matter volume was furthermore compared between groups using the FreeSurfer V6.0 toolbox with default parameters 13 after cortical parcellation according to the Desikan-Killiany atlas (DKT) 14.”*

No smoothing was applied to the macrostructural data as they were only managed in a region-of-interest based approach.

Could the authors also please justify the use of a small Gaussian kernel of 3 mm (esp. when their initial diffusion MRI data is of 3 mm in thickness before upsampling, which should perhaps call for a larger kernel of at least 6 mm)?

The dMRI is sampled anisotropically (1.5,1.5,3) and upsampled to 1.5 isotropic resolution (details are provided within the Methods section on page 32). We consider a smoothing with twice the voxel size of the sampled data appropriate.

The authors also mention applying Bonferroni corrections, but as far as I can see, it is never specified across what, and how many comparisons are then corrected for (how many voxels, parcels/regions, number of clinical variables, etc.).

Voxel-wise comparisons were always corrected with the family-wise error approach. This was also applied to the FreeSurfer-derived metrics. Bonferroni correction was applied for ANCOVAs comparing whole-brain gray and white matter DMI parameters between groups correcting for 6 comparisons (3 (V-extra, V-intra, V-CSF) x 2 (GM and WM)). Furthermore, it was applied to Spearman's rank correlation tests comparing correlations between clinical data (i.e. current disability, disease severity, MoCA-performance, WEIMuS, olfactory performance, and GDS-15; altogether 25 comparisons).

Results page 7: *“Between the parameters current disability, disease severity, MoCA-performance, WEIMuS, olfactory performance, and GDS-15, a significant association was only present between GDS-15 and WEIMuS ($P = 0.012$) after correcting for multiple comparisons.”*

Results page 9: *“A comparison of whole-brain gray and white matter DMI parameters between groups was performed using ANCOVAs controlling for “age” and “sex”, Bonferroni-correction was performed to account for multiple comparisons.”*

Method page 34: *“For ANCOVAs comparing whole-brain gray and white matter DMI parameters and correlations between clinical data, Bonferroni-correction was applied for multiple comparisons.”*

The definition and numbers of variables in the models vary, and as mentioned before, without full justification. In different parts of the text for seemingly the same analyses, the same variables are described in turn as “nuisance covariates” or “covariates” (of interest). Presumably the same linear models were used, and it only depends on whether the authors focused their attentions to the betas/stats associated with one variable or another, but some consistency would be welcome.

We thank the reviewer for this valuable advice. Within the revised version of our manuscript, the covariates' age, sex and disease severity score were termed nuisance covariates.

Finally, age x group interactions are not taken into account at the moment, when it would appear from previous publications that there is a differential effect of age associated with covid. The authors should consider adding this interaction in their model.

As suggested, we assessed the interaction of age and group for total gray matter V-CSF, V-extra and V-intra. However, no significant interaction was observed after Bonferroni corrections for multiple comparisons. Thus, this interaction was not added into our model.

4. Direction of effects. Different parts of the text confusingly refer to presumably the same findings in opposite way. For instance, Figure 2B refers to a significant *decrease* in Vextra in the covid group: “Voxels with significantly decreased V-extra in Post-COVID-syndrome patients were indicated by blue shading”, whereas p9, line 25 of the Results mentions a significant *increase* in Vextra in the covid group: “Areas with significantly increased V-extra in Post-COVID patients encompass widespread cortical (...)”.

Thank you for this thoughtful comment. We have taken great care to avoid this when rewriting the manuscript.

In line with this, when looking at Figure 2A, there appears to be strong, opposite effects (in blue color vs in red color). Have the authors explored the significance of the effects in both directions? In Figure 2B, only those areas corresponding to the blue effects seem to have been investigated. Even if the opposite effects might be down to artifacts, this needs to be explored and discussed.

The reviewer is right, indeed, we noted strong effects in different directions. Thus, we explored the significance of the effects in both directions. Although opposing effect directions with substantial effect sizes were observed in the initial version of the manuscript, significantly different voxels were limited to the GM, with a reduction of V-extra in the COVID-19 cohort compared to the HC.

In the current version of the manuscript, significant voxels now comprise both positive and negative effect directions (see Figure 2 and Results: 4. Spatial distribution and direction of V-extra changes across groups). We have explicitly addressed this phenomenon in detail within the revised manuscript:

Discussion page 13: *“However, voxel-based comparisons of V-extra between groups revealed an even more distinguished view on the COVID-19-related effect: whereas a marked V-extra decrease occurred in neocortical gray matter and thalamus, increased V-extra was present within the corpus callosum, internal capsule, cerebellum and brainstem. However, the PCC- and UPC-group can be distinguished from each other by their different emphasis on this opposite pattern: while cortical V-extra decrease prevailed by PCC-patients, infratentorial V-extra increase was pronounced in UPC-subjects.”*

Discussion page 14: *“PCC-patients are characterized by a volume shift from gray matter V-extra (i.e. the compartment of cell bodies and extracellular matrix) into V-CSF (representing interstitial free fluid and perivascular spaces), that could be explained by shrinkage or even degeneration/loss of cells (10.1016/j.nicl.2017.03.003). ... Although a volume shift from V-extra into V-CSF in cortical gray matter was also observed in UPC-subjects, an opposite process with a volume shift into the compartment of cell bodies and extracellular matrix with emphasis on brainstem and cerebellum prevailed.”*

5. Data availability. I do not know whether a provision has been made with the Editors, but the current approach, namely “Data and code are available from the authors upon reasonable request and approval of the ethics committee” is regrettably inadequate, particularly after checking the NPG requirements: A condition of publication in a Nature Portfolio journal is that *authors are required to make materials, data, code, and associated protocols promptly available to readers without undue qualifications*. Any restrictions on the availability of materials or information must be disclosed to the editors at the time of submission. Any restrictions must also be disclosed in the submitted manuscript.

<https://www.nature.com/nature-portfolio/editorial-policies/reporting-standards>

We followed the reviewer’s advice and added patient data as “Source Data” to the submission. In addition, we revised the manuscript:

Methods page 34: Source data are provided with this paper. Raw MRI data can be obtained from the corresponding author upon reasonable request.

6. Minor issues. “MRI” is sometimes misspelled “MRT” in “ Δ positive PCR - cMRT (days)”. “Gust” is obsolete.

We thank the reviewer for this careful observation and changed the manuscript accordingly.

Reviewer #2 (Remarks to the Author):

Thank you for asking me to look at this paper which makes some interesting imaging observations in relation to Post-Covid Syndrome. When I agreed to review the paper I did not realise quite how involve the imaging studies were, and this is not my area of expertise. I therefore can find myself to some general observations. The main observation in this study was gray matter volume shifts which were compatible with accelerated ageing and comparable to the previous studies. They found these alterations correlated with severity of initial infection but not with clinical outcomes in terms of olfactory performance, fatigue and cognition. They therefore concluded that they were an epiphenomenon of Covid-19 rather than the structural basis of the longer-term complications.

We highly appreciate the reviewer’s comments on our work.

1.) For me there is a flaw in the logic here. Just because A does not directly correlate with B, it does not therefore follow that A cannot have any causal relationship with B. It may be that there are other compounding causal factors involved which have not been measured, and thus not been allowed for. I think they allowed for age and sex, but what about other potential factors like comorbidities, smoking, alcohol use, etc. I think it is worth at least discussing this even if you are not able to study it and allow for it.

We thank the reviewer for these important remarks. As recommended by the reviewers 1 and 2, we enrolled a novel control group (i.e. actually unimpaired patients that passed COVID-19; UPC-group) and substantially expanded the population of patients with Post-COVID-Condition (PCC). Based on these changes, we were able to draw fundamentally novel conclusions about the impact of COVID-19 on the brain and the pathophysiology of PCC: COVID-19 induced a specific pattern of mutual volume shifts between the extraneurite compartment (V-extra, i.e. somata, extracellular matrix) and the free water fraction (V-CSF, i.e. cerebrospinal fluid, perivascular spaces): whereas V-extra decrease occurred in neocortical gray matter and thalamus, increased V-extra was present within corpus callosum, internal capsule, cerebellum and brainstem. Moreover, PCC- and UPC-patients differed regarding their emphasis on this pattern. Correlations between V-extra and clinical scores revealed affection of symptom-specific networks associated with impaired cognition, or olfactory-performance and fatigue.

In the light of the new analyses, this study now revealed cerebral microstructural changes after COVID-19 with distinguishable patterns in patients with and without PCC. PCC-symptoms were associated with affection of specific networks, providing a biomarker and pathophysiological basis of this syndrome.

2.) Secondly they put out in the discussion that it would've been helpful to have a control group of previous Covid 19 patients who do not now have postcovid syndrome, but in whom they perform the same imaging studies. I would have thought this was essential to do to support their argument that the structural changes they see are simply an effect of severe Covid and have no relationship with the long-term post Covid problems. I think the fact that they did not include any controls at all it's regrettable it significantly we can see the paper.

We thank the reviewer for this valuable advice! As suggested and as stated in the answer to point 1, we enrolled a novel control group comprising individuals that passed COVID-19 without having current complaints (n=38).

I am sorry that I am only able to provide this limited report, but detailed imaging studies are really beyond my area of expertise.

We are very grateful for these valuable thoughts on our data, which have helped us improve the methodology and provide groundbreaking insights into the impact of COVID 19 infection on the central nervous system.

Reviewer #3 (Remarks to the Author):

The manuscript presents brain tissue microstructure changes, obtained with diffusion MRI, in gray matter in patients with post-Covid syndrome. The work adds to the on-going debate about the microstructural (cellular-level) origins of the syndrome, and specifically stipulates that the recently found association between "free water" fraction increase and the disease does not imply that the post-Covid syndrome originates due to such changes. While I find the results intriguing and timely, I have a few suggestions.

We thank the reviewer for his comments and suggestions for improvement, which greatly contributed to the revision of our data.

1. The major one is about assigning of model's "compartments" to intra- and extra-neurite. The dMRI "Standard Model" (SM) the authors are using has been developed for white matter, as it implies no exchange between the one-dimensional compartment (originally ascribed to axons) and the extra-axonal space. (The third compartment, CSF, corresponds to macroscopic partial-volume inclusions in a voxel.) While SM has been previously applied to gray matter, lumping together axons and dendrites into a "neurite" compartment non-exchanging with extra-cellular space has never been justified. In fact, recent studies have been pointing at a non-negligible exchange between dendrites and extra-cellular space, on the scale of ~10ms (Williamson eLife 2019), 20-60ms (Jelescu Neuroimage 2022), or even <10ms (Olesen Neuroimage 2021). In any case, the exchange time seems to be smaller or of the order of the diffusion time in clinical dMRI. Not surprisingly, the intra-neurite volume fraction the authors obtain is < 25%, much below that of neuropil in gray matter. It may as well be that this fraction corresponds mostly to axons (that are inevitably present in the voxels due to partial-volume effects), although the dendrites may somewhat affect this fraction depending on the exchange rate that, alas, has still not been determined reliably. While I do not think this issue should disqualify the work, the authors should be more careful about interpreting their results in terms of specific compartments, and a discussion about exchange and applicability of SM-like models without exchange is required.

We thank the reviewer very much for these thoughtful considerations that definitely require further discussion. We furthermore agree that we cannot sharply distinguish between V-intra (neurites) and V-extra (cells and extracellular matrix). This also explains that the intra-neurite volume fraction we obtain in gray matter is rather low (compared to the fraction of neuropil present in gray matter). So, the present finding may be rather a shift from (V-intra + V-extra) towards V-CSF. However, these considerations are still compatible with our hypothesis of an accelerated aging process.

We addressed these issues in the limitation section of our Discussion (page 18): *"In addition to its cross-sectional design, following limitations have to be considered for the interpretation of our present study. The dMRI "standard model" used by DMI has been developed for white matter, as it implies no exchange between the one-dimensional compartment (V-intra) and the extra-axonal (V-extra) and CSF space. While this assumption is well justified by the myelin-sheaths in white matter, it might not be valid in gray matter. In fact, recent studies have been pointing at a non-negligible exchange between dendrites and extracellular space, on the scale of ~10ms (30735128), 20-60ms (35523369), or even <10ms (33582270). Here, we cannot sharply distinguish between V-intra (neurites) and V-extra (cells and extracellular matrix). This also explains that the intra-neurite volume fraction we obtain in gray matter is rather low (compared to the fraction of neuropil present in gray matter). So, the present finding is rather a shift from (V-intra + V-extra) towards V-CSF. However, these considerations are still compatible with our aforementioned hypothesis of an accelerated aging process."*

Other issues:

1. Placing the work in context: DMI is mentioned from the very beginning, yet it's referenced somewhere in the middle (refs 12,13), with particular refs being somewhat tangential to this work (ref 12 is on tractography, and ref 13 is about diffusion time-dependent effects beyond the adopted SM picture). I suggest referring to recent review(s) on the subject. Furthermore, tissue microstructure imaging with dMRI relies on many models, whereas the authors adopt a particular one, the SM (defined above). Its brief description, assumptions (ie Gaussian diffusion and absence of exchange), and key SM references, should be part of the main text, as the use of this model underpins all the interpretations.

We thank the reviewer for this thoughtful observation which we implemented in the actualized version of the manuscript. Now, the DMI technique and the standard model are referenced much earlier in the Introduction on page 6. In addition, we referred to Novikov et al. 2018 as a recent review on the subject. As stated before and suggested by the reviewer, we included more information on the SM to the limitations section as stated in the answer to the main comment and included more key references on the SM to the Methods section on page 33 (Jespersen et al., 2007, Novikov et al., 2018, Reisert et al., 2017, Zhang et al., 2012).

2. Sensitivity to volume fractions: Please show results of noise propagation with the SNR you have in your data, for the specific protocol, and the parameters (volume fractions) that are highlighted in the work. Can you quantify the spurious correlations for the two independent volume fractions inevitable due to the nonlinear character of the model and the degeneracies in its parameter estimation? This is important since it's known that estimating the free water fraction is particularly difficult in a limited 2-shell protocol.

In fact, the CSF fraction is the most stable parameter within the model. For this, we refer to Reisert et al. 2017 where a protocol very similar to the one used in this study was already analyzed. Assuming an SNR of 20, the error of the V-CSF is in the range of 0.1. Due to the nonlinear nature of the model it is indeed difficult to quantify the correlations. Depending on the diffusivities of the compartments, their discrimination becomes less or more robust.

3. Protocol: it's stated that there are 65 diffusion directions, yet 58 images per shell. Should there be 65 images per shell then?

The reviewer is right and we accordingly corrected this mistake. The current version of the manuscript includes on page 32:

The diffusion weighted sequence was acquired with the following parameters: axial orientation, 42 slices, voxel size 1.5 x 1.5 x 3 mm³, TR 2800 ms, TE 88 ms, bandwidth 1778 Hz, flip angle 90°, simultaneous multi-band acceleration factor 2, GRAPPA factor 2, 58 diffusion-encoding gradient

directions with b-factors 1000 and 2000 s/mm², 15 non-diffusion weighted images (interleaved during diffusion-encoding directions).

4. Can you comment about the validity of upsampling 3mm to 1.5mm in a very narrow cortex?

Even at 1.5 mm isotropic raw data one cannot exclude a certain extent of partial volume effects to the cortex band. The upsampling of the 1.5x1.5x3 mm to 1.5 mm isotropic voxel size is justifiable in our opinion, since we applied smoothing of 3 mm FWHM for the voxel-wise group comparisons (please see our comment on reviewer 1).

REVIEWER COMMENTS

Reviewer #1 (Remarks to the Author):

The authors have to be commended for not only increasing the sample size of their main group of interest, but also for adding a new control group, with people who had been infected with SARS-CoV-2, but do not suffer from long COVID. This in effect produces an entirely new study, with much more compelling, and more interpretable findings.

A few major and minor concerns, easily addressable, remain.

1. At present, the healthy controls' demographics are in a separate Supplementary Table 1. These should appear in the main Table 1, and statistically compared in that Table with both COVID groups in the same way that both COVID groups are compared with one another, esp. with respect to their risk factors/comorbidities (if some of the comorbidities information is not available, please state so in Table 1).

There is also no information on how these healthy controls are confirmed not to have had COVID. The only mention seems to be "controls without a history of COVID-19" – is this based on their medical records, on self-report, or...? This information should be clearly stated in the manuscript.

A discussion on the (probable) lack of information on key variables such as SES and certain risk factors (e.g., smoking, blood pressure, etc.) for the controls and the two COVID groups, as well as significant differences between these 3 groups, and the possibility that some controls have been infected with COVID (if indeed this has not been verified with an antigen test), is also needed.

2. Even if the time between PCR and MRI is not significantly different between the two COVID groups, it would be interesting to investigate the effect of this period of time per se, both (a) within group: do the GM abnormalities (e.g., obtained by contrasting long COVID vs healthy controls) seem to get better/worse over time in the long COVID group?; and (b) across the two COVID groups: is this effect different between the recovered and long COVID groups?

3. Minor comment: upsampling 3mm-thick voxels to 1.5mm only to smooth them using a 3mm Gaussian kernel does not achieve the smoothing the authors are after along the z-axis. "Twice the voxel size" is indeed the usual rule as the authors state, but upsampling their voxels along z does not create any information, and the resolution of the data is still, in effect, of 3mm in z.

It would be far too much work to redo the analyses, but it would have been good if, at least, the authors had acknowledged this rather low resolution as a clear limitation, as indeed the fact that their original data was acquired using anisotropic voxels, which is less than optimal (see also their - insufficient in my view - response to Point 4 of Reviewer 3).

4. Regarding correction for multiple comparisons: the authors state that this has been done for every analysis. They give in their responses the details of what they have corrected across, but this information is still missing for the manuscript.

Please explicitly mention the information, at least in the Methods section, about what the correction has been applied across - e.g., across voxels, across ROIs, across 6 comparisons - 3 (Vextra, V-intra, V-CSF) x 2 (GM and WM), across 25 clinical scores, etc.

5. Could the authors please be more explicit in their text as to what they mean by "Source Data" that is made available with the manuscript?

6. Could the authors please specify in the captions of Figures 2 and 3 if the P-values for their displayed

voxel-wise results are corrected for multiple comparisons?

7. The manuscript would perhaps benefit from being proofread by a native speaker (or very fluent English speaker). Some wording can be confusing or misleading to the reader, in particular (and please correct the summary and abstract accordingly):

- "Patients that passed COVID-19". The authors presumably mean "patients with past COVID-19", or "patients that had been infected with SARS-COV-2", or "patients that had contracted COVID-19"
- A variable does not get "implant(ed)" in a model (although I do get the meaning!), it gets e.g., "added".
- "Voxel-based comparisons of V-extra between groups however display an even more distinguished view on the COVID-19-related effect": "distinguished" means illustrious, or eminent. If you mean "that can help distinguish": it will be "distinct" or "distinguishable".

Reviewer #2 (Remarks to the Author):

The authors have now increased their number of subjects (by about 40 percent) and included the suggested mild covid-19 control group (albeit a small group of 38 subjects). By including this group, and doing some of the more detailed and thorough analyses suggested, the key messages of the study have fundamentally changed. For me therefore this completely discredits the original submission, and therefore the team that submitted it. Presumably they thought they had a robust manuscript with strong messages at the first submission. Now they include the control group that should have been there in the first place, they come up with a completely different message. Early in the pandemic when everything was new, I think the community was willing to be forgiving. But we are now a couple of years into the pandemic we really must aim for more robust science if we are going to try and better understand the disease mechanisms and impacts of Covid-19 infection. For me that requires inclusion of an obvious control group in studies from the start, and it requires manuscripts to have a robust and consistent message, not one which fundamentally changes when an obvious control group is included. Peer review has a vital role in strengthening and improving a manuscript, and perhaps modifying the message slightly. But if peer review results in a completely different paper, then I think we have to question the validity of the whole submission.

I hope this is helpful.

Reviewer #3 (Remarks to the Author):

The manuscript has improved and I believe it can be published, as long as a few minor issues are addressed.

1. Regarding the stability of CSF fraction, it depends on the range. Indeed, when "coarsely" looking at the whole range $v_{csf} = 0 - 1$, its determination looks indeed quite stable. However, as your own noise propagation shows in 2017, at low $v_{csf} \sim 0.1$, at the bottom-left corners of your plots, the estimation has $\sim 50\%$ errors, depending on the protocol. Yet this is the actual range of this fraction for most voxels, and also for your $v_{csf} \sim 0.15$ in Fig.1. We have recently studied noise propagation of v_{csf} in detail for limited protocols, and our general conclusion is similar, for small v_{csf} the *relative* errors are quite large.
2. The added Ref 47 Williamson is a wrong Williamson (not from Basser group). Use Williamson et al. eLife 2019;8:e51101. DOI: <https://doi.org/10.7554/eLife.51101>.
3. Ref 59 (the review of multi-compartment models, where the term "Standard model" has been

introduced) should accompany either Refs 8 and 9, or the part in Discussion where "standard model" is mentioned (whereas Ref 8 is not about multi-shell but about microstructure footprint in the time-dependence).

4. Protocol: "58 diffusion-encoding gradient directions with b-factors 1000 and 2000" — it's unclear how many of 58 directions belong to $b=1000$ and how many to $b=2000$, or is it 58 directions per each shell?

Authors' Response to Reviewers Comments:

Manuscript Number: NCOMMS-22-15714A-Z

MS TITLE: Cerebral microstructural alterations in Post-COVID-condition are related to cognitive impairment, olfactory dysfunction and fatigue

We would like to thank the reviewers for their thorough review and valuable comments on our manuscript. In the following, we provided a point-by-point response to the individual comments (answers are given in red, and changes to the manuscript in *italics*). In the revised manuscript file, changes are highlighted in yellow. Furthermore, the manuscript underwent language editing, which is highlighted in blue.

Reviewer 1:

The authors have to be commended for not only increasing the sample size of their main group of interest, but also for adding a new control group, with people who had been infected with SARS-CoV-2, but do not suffer from long COVID. This in effect produces an entirely new study, with much more compelling, and more interpretable findings.

We thank the reviewer for this positive evaluation of our work and the helpful suggestions to further improve our manuscript.

A few major and minor concerns, easily addressable, remain.

1. At present, the healthy controls' demographics are in a separate Supplementary Table 1. These should appear in the main Table 1, and statistically compared in that Table with both COVID groups in the same way that both COVID groups are compared with one another, esp. with respect to their risk factors/comorbidities (if some of the comorbidities information is not available, please state so in Table 1).

In response to this suggestion, we transferred the respective information to **Table 1** and carried out group comparisons. We subsequently noted significant differences regarding adipositas, arterial hypertension, migraine and obstructive sleep apnoea, to the detriment of PCC-patients.

We added this information within the **Results** section of the revised manuscript at page: "*Analysis of comorbidity occurrence revealed significant differences between groups (PCC,*

*UPC and HNC) in terms of adipositas, arterial hypertension, migraine and obstructive sleep apnoea, to the detriment of PCC patients (see **Table 1**).“*

As this higher proportion of subjects with comorbidities in the PCC-group is a potential confounder, we mentioned and discussed this in the limitation section of the **Discussion 19**: *“Although a significantly higher proportion of subjects in the PCC cohort suffered from migraine or had arterial hypertension and obesity as risk factors for impaired brain integrity, it is unlikely that the patterns we observed were due to these factors, as these pathological changes impact the brain differently to what we observed⁵²⁻⁵⁴.”*

2. There is also no information on how these healthy controls are confirmed not to have had COVID. The only mention seems to be “controls without a history of COVID-19” – is this based on their medical records, on self-report, or...? This information should be clearly stated in the manuscript.

The HNC-group was enrolled as healthy controls without evidence of a previous COVID-19 infection as obtained by self-report and all available medical records. However, as have not determined the antigen status of HNC-subjects, we cannot ultimately exclude the possibility that some of the HNC group were infected with COVID-19 prior to enrollment.

We added this information on this to the **Methods** section at page 33:” 2) *“Healthy non-COVID” (HNC group): an in-house collective of 46 healthy subjects (age: 44 [31]; range: 21 to 80 years; 23/23 males/females – self-reported), with no history of COVID-19 infection (obtained from medical records and self-reports) and no significant difference in age to subjects in the PCC and UPC groups (Mann-Whitney-U, $P = 0.22$).”*

Moreover, we now address this in the limitations section of the **Discussion** at page 19: *“Although the HNC group was enrolled without history of a past COVID-19 infection, we cannot rule out with certainty that some subjects had previously contracted an infection, as antigen status was lacking.”*

3. A discussion on the (probable) lack of information on key variables such as SES and certain risk factors (e.g., smoking, blood pressure, etc.) for the controls and the two COVID groups, as well as significant differences between these 3 groups, and the possibility that some controls

have been infected with COVID (if indeed this has not been verified with an antigen test), is also needed.

Unfortunately, we did not obtain these key variables during the participants' workup and examinations. Moreover, we cannot ultimately exclude the possibility that some of the HNC group were infected with COVID-19 prior to enrollment.

This was added to the limitations section of the **Discussion** in the revised version of the manuscript at page 19: *“The soundness of our results could have been further strengthened by considering key variables such as socioeconomic status or risk factors of cerebral microstructural integrity (e.g. smoking status), which unfortunately were not available in all cohorts.”*

4. Even if the time between PCR and MRI is not significantly different between the two COVID groups, it would be interesting to investigate the effect of this period of time per se, both (a) within group: do the GM abnormalities (e.g., obtained by contrasting long COVID vs healthy controls) seem to get better/worse over time in the long COVID group?; and (b) across the two COVID groups: is this effect different between the recovered and long COVID groups?

We highly appreciated this suggestion. Accordingly, we assessed the impact of the delay between the infection and the MRI scan. At first, we investigated the relationship between the time period between the first positive COVID-19 test and imaging with V-extra using linear regression for PCC, UPC as well as PCC and UPC combined. Two different approaches were applied for V-extra extraction: 1) by extracting V-extra from the whole GM. 2) by extracting V-extra from a mask obtained by contrasting PCC vs. HC for V-extra to capture the area of the maximal effect (see **Review-specific Figure 1**).

Review-specific Figure 1: Binary mask of the significant results from voxel-wise comparisons of V-extra after threshold-free cluster enhancement and FWE-correction between the PCC- and HNC-groups superimposed onto a T1w MRI template.

No significant associations were observed for the PCC group [whole GM: (beta =0.006544, $p = 0.784$); mask: (beta =-0.03122, $p = 0.158$)], the UPC group [whole GM: (beta =-0.01904, $p = 0.215$); mask: (beta =-0.01025, $p = 0.431$)] and UPC+PCC combined [whole GM: (beta = 0.002527, $p = 0.867$); mask (beta =0.003566, $p = 0.819$)].

Furthermore, we added the delay between the infection and the MRI scan as a covariate into our voxel-wise model investigating comparisons of V-extra between the PCC- and UPC-groups in addition to the covariates “age” and “sex”. Here, adding “delay” did not change the spatial distribution of significant V-extra changes between the PCC- and the UPC-group (**Review-specific Figure 2**).

Covariates “age“ and “sex“

Covariates “age“, “sex“ and “delay“

Review-specific Figure 2: Significant results from voxel-wise comparisons of V-extra after threshold-free cluster enhancement and FWE-correction between the PCC- and UPC-groups, superimposed onto a T1w MRI template. The upper panel of results were obtained with the covariates “age” and “sex”, whereas the results in the lower panel used “age”, “sex”, and “delay between infection and MRI” as covariates. Radiological orientation: left side of the image corresponds to the patient’s right side; numbers denote the axial (z) position in millimeters.

Thus, we found no effect of the delay between positive PCR and MRI scan and V-extra within and across different groups. At this stage, one may speculate if V-extra changes are reversible or not. However, due to the cross-sectional design of our study, we prefer to be very cautious with any conclusions. Longitudinal studies are required to make definite statements on reversibility of microstructural changes after COVID-19 and in PCC patients.

5. Minor comment: upsampling 3mm-thick voxels to 1.5mm only to smooth them using a 3mm Gaussian kernel does not achieve the smoothing the authors are after along the z-axis. “Twice

the voxel size” is indeed the usual rule as the authors state, but upsampling their voxels along z does not create any information, and the resolution of the data is still, in effect, of 3mm in z. It would be far too much work to redo the analyses, but it would have been good if, at least, the authors had acknowledged this rather low resolution as a clear limitation, as indeed the fact that their original data was acquired using anisotropic voxels, which is less than optimal (see also their - insufficient in my view - response to Point 4 of Reviewer 3).

We appreciate this suggestion and now discuss this limitation in the Discussion of the revised manuscript at page 19: *“Furthermore, the sampling of the diffusion-weighted sequence with anisotropic voxels with a resolution of 3 mm in the z-direction might introduce a potential bias. However, due to the rather global nature of the present finding, this is unlikely to have substantially hampered the results.”*

4. Regarding correction for multiple comparisons: the authors state that this has been done for every analysis. They give in their responses the details of what they have corrected across, but this information is still missing for the manuscript. Please explicitly mention the information, at least in the Methods section, about what the correction has been applied across - e.g., across voxels, across ROIs, across 6 comparisons - 3 (Vextra, V-intra, V-CSF) x 2 (GM and WM), across 25 clinical scores, etc.

We now explicitly mention the information within the **Methods** section of the revised manuscript.

Methods section, page 35: *“As implemented in the Statistical Parametric Mapping-Voxel-Based Morphometry (SPM-VBM) 8-Toolbox, voxel-based group comparison of whole-brain DMI parameters was performed using a parametric multiple regression model ... The family-wise error (FWE)-method was employed to correct for multiple comparisons (i.e. across voxels).“*

Methods section, page 35: *“As Cortical morphometry including cortical thickness, cortical surface area ... and the FWE method was applied to correct for multiple comparisons (i.e. across DKT-ROIs).“*

Methods section, page 35: “*To correlate clinical outcomes with DMI parameters ... voxel-based analyses were performed ... FWE-correction was applied to control for multiple comparisons (i.e. across voxels).*”

Methods section, page 36: “*Correlations between clinical data (i.e. current disability, disease severity, MoCA-performance, WEIMuS, olfactory performance, and GDS-15) were identified using Spearman’s rank correlation test, where Bonferroni-correction has been applied across 15 comparisons. ... For ANCOVAs comparing whole-brain gray and white matter DMI parameters, Bonferroni-correction was applied to account for multiple comparisons as follows: For whole-brain gray and white matter DMI parameters, correction has been applied across 6 comparisons (3 [V-extra, V-intra, V-CSF] x 2 [white and gray matter]). In the whole-brain DMI-analysis that further accounted for disease severity, correction has been applied across 12 comparisons (3 [V-extra, V-intra, V-CSF] x 2 [white and gray matter] x 2 [effect of group and disease severity]).*”

By describing these specifications in detail, we realized that we applied the Bonferroni-correction for the whole-brain DMI-analysis that further accounted for disease severity for only 6 instead of 12 comparisons. We apologize for this and implemented the corrected version into the **Results** section of the revised manuscript. However, these corrections did not lead to different statements.

Results section, page 9: “*While disease severity contributed significantly to the decrease in gray-matter V-extra ($P = 0.007$; $t = -3.51$) and increase in gray-matter V-CSF ($P = 0.016$, $t = 3.28$), the group-effect still remained significant for V-extra ($P = 0.008$; $df: 168$; $t = 3.48$).*”

5. Could the authors please be more explicit in their text as to what they mean by “Source Data” that is made available with the manuscript?

Anonymized numeric data will be made available in a public repository via <https://doi.org/10.5061/dryad.kkwh70s9g>. This link will be valid as soon as the paper becomes published. For the review process, the datasheet will be shared with the reviewers as a supplementary file. As the raw MRI data may contain information that could compromise the participants’ privacy, they can only be made available on reasonable request from the corresponding author (JAH). The code used in this study is publicly available via <https://bitbucket.org/reisert/baydiff/wiki/Home>, <https://github.com/spisakt/pTFCE>,

<https://www.fil.ion.ucl.ac.uk/spm/software/download/> and
<https://doi.org/10.5061/dryad.kkwh70s9g>. This is now clearly indicated in the Data and Code availability statements in the revised manuscript at page 37.

6. Could the authors please specify in the captions of Figures 2 and 3 if the P-values for their displayed voxel-wise results are corrected for multiple comparisons?

P-values were corrected for multiple comparisons using the Family-wise error rate (FWE), with “age” and “sex” as nuisance covariates. We implemented this information into the captions of Figure 2 and 3.

Caption **Figure 2** at page 31: *“Indicated P-values are corrected for multiple comparisons across voxels using the Family-wise error rate (FWE) with nuisance covariates “age” and “sex”.”*

Caption **Figure 3** at page 33: *“Accordingly, P-values were corrected for multiple comparisons across voxels using the family-wise error rate (FWE) and “age” and “sex” as nuisance covariates.”*

7. The manuscript would perhaps benefit from being proofread by a native speaker (or very fluent English speaker). Some wording can be confusing or misleading to the reader, in particular (and please correct the summary and abstract accordingly):

Following the reviewer’s suggestion, a native speaker has now proofread our manuscript and the corrections have been incorporated into the revised version of our manuscript and have been in **blue**.

- “Patients that passed COVID-19”. The authors presumably mean “patients with past COVID-19”, or “patients that had been infected with SARS-COV-2”, or “patients that had contracted COVID-19”

We have rephrased this throughout the manuscript.

- A variable does not get “implant(ed)” in a model (although I do get the meaning!), it gets e.g., “added”.

We thank the reviewer for pointing this out and have changed the sentence in the **Results** section, page 11 accordingly: *“Although “disease severity” significantly affected whole gray*

matter V-extra (see above), adding it into the model as nuisance covariate did not change the spatial distribution of the found results (Supplementary Figure 1).”

- “Voxel-based comparisons of V-extra between groups however display an even more distinguished view on the COVID-19-related effect”: “distinguished” means illustrious, or eminent. If you mean “that can help distinguish”: it will be “distinct” or “distinguishable”.

We have rephrased this throughout the manuscript.

Reviewer 2:

The authors have now increased their number of subjects (by about 40 percent) and included the suggested mild covid-19 control group (albeit a small group of 38 subjects). By including this group, and doing some of the more detailed and thorough analyses suggested, the key messages of the study have fundamentally changed. For me therefore this completely discredits the original submission, and therefore the team that submitted it. Presumably they thought they had a robust manuscript with strong messages at the first submission. Now they include the control group that should have been there in the first place, they come up with a completely different message. Early in the pandemic when everything was new, I think the community was willing to be forgiving. But we are now a couple of years into the pandemic we really must aim for more robust science if we are going to try and better understand the disease mechanisms and impacts of Covid-19 infection. For me that requires inclusion of an obvious control group in studies from the start, and it requires manuscripts to have a robust and consistent message, not one which fundamentally changes when an obvious control group is included. Peer review has a vital role in strengthening and improving a manuscript, and perhaps modifying the message slightly. But if peer review results in a completely different paper, then I think we have to question the validity of the whole submission. I hope this is helpful.

In principle, we understand the concern raised by Reviewer 2 and recognise that every manuscript must be scrutinized very critically in the review process. However, the peer review process and the reviewer’s suggestions led to a substantial improvement of the methodology and the overall soundness of our research. As part of this, the expansion of our cohort and enrollment of an additional control group did alter the interpretation of our results.

Reviewer 3:

The manuscript has improved and I believe it can be published, as long as a few minor issues are addressed.

We thank the reviewer for this positive evaluation of our work.

1. I would suggest to the authors to be less categorical in their claims about V-extra, in the view of the discussion about the role of exchange (see my comment and their response regarding the validity of Standard Model in gray matter). While in the abstract they describe the shifts mostly with respect to V-extra, they do acknowledge that possible water exchange between intra- and extra-neurite compartments blurs their distinction: "So, the present finding is rather a shift from (V-intra + V-extra) towards V-CSF" (p.18, lines 407-408 of revised MS). I agree with this interpretation, and to be on a safer side, would recommend them make this distinction early in the main text rather than in Methods, and, whenever possible, make the statements in terms of "free water" V-CSF changes, as it would be more reliable."

We agree that a more defensive interpretation of our results better addresses the fact that we employed a technique based on the Standard Model to investigate gray matter. We carefully revised the whole manuscript and rephrased the statements as suggested. Moreover, we introduced the term "membrane-enclosed compartment" to describe the combination of V-extra and V-intra, and used this whenever appropriate.

In the **Summary paragraph** at page 2: *"Here, COVID-19 induced a specific pattern of mutual volume shifts between the membrane-enclosed compartment (i.e. somata, neurites, and extracellular matrix) and the free water fraction (V-CSF, i.e. cerebrospinal fluid, perivascular spaces): Whereas a decrease in the volume of membrane-enclosed compartment occurred in neocortical gray matter and thalamus, an increase was present within the corpus callosum, internal capsule, cerebellum and brainstem. Moreover, PCC- and UPC-patients differed in terms of the distribution of this pattern."*

In the **Abstract** at page 2: *"Analysis of whole-brain DMI-data revealed a volume shift from the membrane-enclosed compartment into the free-water fraction in the gray matter, which was positively associated with the severity of initial COVID-19 infection ($P = 0.007$). However, voxel-based inter-group comparisons of DMI parameters allowed an even more distinguishable view of the COVID-19-related effect: Whereas a marked decrease in the volume of the membrane-enclosed compartment occurred in neocortical gray matter and*

thalamus, an increase was observed within the corpus callosum, internal capsule, cerebellum and brainstem.”

In the Results at page 9: *“It is also worth noting that this biophysically motivated approach was initially developed for the white matter, which implies a less stable differentiation between V-intra and V-extra in gray matter ⁹. Nevertheless, our findings suggest a shift from the membrane-enclosed compartment (V-intra + V-extra) towards V-CSF. ... we restricted further analyses to the parameter V-extra, not least because we observed the largest effects on this parameter and the neurite fraction V-intra is rather small within the gray matter.”*

In the Discussion at page 13-14: *“On a gross scale, a significant volume shift from the membrane-enclosed compartment (i.e. V-extra + V-intra with predominant effects on V-extra) into the microstructural free-water compartment (V-CSF) occurred in the gray matter of PCC patients and correlated with initial disease severity. ... The dMRI “standard model” on which DMI is based was originally developed for white matter, as it implies no exchange between the one-dimensional compartment (V-intra) and the extra-axonal (V-extra) and CSF space ^{9,10}. While this assumption is validated by the presence of myelin sheaths in white matter, it might not be applicable to gray matter. In fact, recent studies have suggested a non-negligible exchange between dendrites and extracellular space on a scale of ~10ms ¹⁹, 20-60ms ²⁰, or even <10ms ²¹. Such minute values do not allow a clear distinction between V-intra (neurites) and V-extra (cells and extracellular matrix) in gray matter. This also explains why the intra-neurite volume fraction we obtained in gray matter was rather low (compared to the fraction of neuropil present in gray matter). Therefore, the present finding more likely represents a shift in volume from the membrane-enclosed compartment (V-intra + V-extra) towards V-CSF. ... In the present study, PCC patients were characterized by shift in volume from the membrane-enclosed compartment of gray matter (i.e. the compartment of cell bodies, neurites and extracellular matrix) to the V-CSF (representing interstitial free fluid and perivascular spaces), which could be explained by shrinkage due to degeneration or cell loss ²²“*

2. Regarding the stability of CSF fraction, it depends on the range. Indeed, when "coarsely" looking at the whole range $v_{\text{csf}} = 0 - 1$, its determination looks indeed quite stable. However, as your own noise propagation shows in 2017, at low $v_{\text{csf}} \sim 0.1$, at the bottom-left corners of your plots, the estimation has ~50% errors, depending on the protocol. Yet this is the actual range of this fraction for most voxels, and also for your $v_{\text{csf}} \sim 0.15$ in Fig.1. We have recently

studied noise propagation of v_{csf} in detail for limited protocols, and our general conclusion is similar, for small v_{csf} the *relative* errors are quite large.

Yes, this is true, the errors for fractions close to 0 or close to 1 get larger (relatively for a small fraction very large) and most of the V-CSF values in white matter are rather low. Fortunately, the range of our findings is more towards 0.2.

We mention this now in the limitations section of the **Discussion** at page 18: *“Regarding the microstructure imaging approach, it should be taken into account that the range of V-CSF values (~0.15) where we found changes is slightly more susceptible to dMRI-related noise than other ranges, where the fractions of V-CSF and V-intra+V-extra are more balanced^{9,51}”*

3. The added Ref 47 Williamson is a wrong Williamson (not from Basser group). Use Williamson et al. eLife 2019;8:e51101. DOI: <https://doi.org/10.7554/eLife.51101>.

Thank you for pointing this out - we have corrected the reference accordingly.

4. Ref 59 (the review of multi-compartment models, where the term "Standard model" has been introduced) should accompany either Refs 8 and 9, or the part in Discussion where "standard model" is mentioned (whereas Ref 8 is not about multi-shell but about microstructure footprint in the time-dependence).

We appreciate this observation and have added this reference where appropriate.

5. Protocol: "58 diffusion-encoding gradient directions with b-factors 1000 and 2000" — it's unclear how many of 58 directions belong to $b=1000$ and how many to $b=2000$, or is it 58 directions per each shell?

Thank you for pointing this out - we have revised this description in the **Methods** section at page 34 as follows: *“(.) The diffusion weighted sequence was acquired with the following parameters: axial orientation, 42 slices, voxel size 1.5 x 1.5 x 3 mm³, TR 2800 ms, TE 88 ms, bandwidth 1778 Hz, flip angle 90°, simultaneous multi-band acceleration factor 2, GRAPPA factor 2, 58 diffusion-encoding gradient directions per shell with b-factors 1000 and 2000 s/mm², and 15 non-diffusion weighted images (interleaved during diffusion-encoding directions); this resulted in a total of 131 images.”*

REVIEWER COMMENTS

Reviewer #1 (Remarks to the Author):

The authors have addressed my comments adequately.

Re 4. and delay between infection and MRI scan, it would be good if the second Figure generated for this answer ('Review-specific Figure 2') were added as a Supplementary Figure, and the two ancillary analyses - namely looking at the effects of delay, and at adding it as a confound - were added to the main ms if not done so already, and discussed with caution (I agree that there are limitations in looking at cross-sectional data).

Re a point raised by the Editor, the long covid cohort was acquired over two distinct periods (June 2020 - Jan 2022 = period 1; Aug 2022 - Oct 2022 = period 2), while the infected cohort without long covid was acquired over the second period only, and presumably the control cohort only during the first period - something that needs to be explicitly stated in the Methods. It would be great if an additional secondary analysis could be carried out, this time adding the period when each participant was acquired as a discrete confounder, on top of sex and age.

Reviewer #3 (Remarks to the Author):

The authors satisfactorily addressed reviewers' comments, I have nothing further to suggest. The manuscript can be published in its present form.

*SPECIFIC REQUEST:

You will see from the Methods section that the PCC sample was recruited in two separate time periods and then combined. It would be helpful to us if you could also comment specifically on whether you have any concerns with this decision, especially in relation to possible inflation of Type 1 errors.

I am not worried about ~7 month lapse in recruitment of PCC sample between January and August 2022, given the overall duration of the recruitment process (6/2020 - 10/2022). Authors also addressed a related concern #4 of Reviewer 1 about delay between infection and MRI.

Authors' Response to Reviewers Comments:

Manuscript Number: NCOMMS-22-15714C

MS TITLE: Cerebral microstructural alterations in Post-COVID-condition are related to cognitive impairment, olfactory dysfunction and fatigue

We again thank the reviewers for their thorough review and their valuable comments on our manuscript. In the following, we provide a point-by-point reply describing how we addressed the individual comments (answers are reported in red, and changes to the manuscript in *italic red*).

Reviewer #1 (Remarks to the Author):

The authors have addressed my comments adequately.

We appreciate this positive feedback!

Re 4. and delay between infection and MRI scan, it would be good if the second Figure generated for this answer ('Review-specific Figure 2') were added as a Supplementary Figure, and the two ancillary analyses - namely looking at the effects of delay, and at adding it as a confound - were added to the main ms if not done so already, and discussed with caution (I agree that there are limitations in looking at cross-sectional data).

As the reviewer suggests, we added the analysis of the "delay" as a confounder and included the 'Review-specific Figure 2' as the novel Supplementary Figure 2 into the revised manuscript. Furthermore, we added a cautious discussion of these findings.

Results section, page 9: *"In contrast, the delay between positive SARS-CoV-2 PCR and MRI-scan did not significantly contribute to changes in gray-matter DMI parameters between the PCC and UPC groups when added as a nuisance covariate together with "age" and "sex" (all $P > 0.05$)."*

Results section, page 11: *"Likewise, adding the "delay" between positive SARS-CoV-2 PCR and MRI scan as a nuisance covariate did not influence the spatial distribution of significant V-extra changes between the PCC- and UPC-group (**Supplementary Figure 2**)."*

Discussion section, page 15: *"Interestingly, the time span between positive SARS-CoV-2 PCR and the cerebral MRI did not influence alterations of gray-matter DMI parameters or spatial distribution of V-extra changes between the PCC and UPC groups. Thus, one could*

speculate on a slow or even non-reversibility of microstructural changes observed here. In line with this assumed chronicity, 85% of patients reporting complaints two months after COVID-19 still reported symptoms one year after their symptom onset¹. However, longitudinal studies are required to make definite statements on reversibility of microstructural alterations after COVID-19 in general and in PCC patients.”

1. Tran, V.-T., Porcher, R., Pane, I. & Ravaud, P. Course of post COVID-19 disease symptoms over time in the ComPaRe long COVID prospective e-cohort. *Nat. Commun.* **13**, 1812 (2022).

Supplementary Figure section, page 14:

Post-COVID-Condition vs. Unimpaired Post-COVID

Supplementary Figure 2. Implementation of the “delay” in days between positive SARS-CoV-2 and MRI-scan into the model as a nuisance covariate did not change the spatial distribution of significant V-extra changes between the PCC- and the UPC-group.

Re a point raised by the Editor, the long covid cohort was acquired over two distinct periods (June 2020 - Jan 2022 = period 1; Aug 2022 - Oct 2022 = period 2), while the infected cohort without long covid was acquired over the second period only, and presumably the control cohort only during the first period - something that needs to be explicitly stated in the Methods.

This fact and the acquisition periods of particular groups are explicitly stated in the Method section of the revised version of our manuscript:

Methods section, page 35: “The PCC patients were recruited in two periods ($n = 62$ between June 2020 and January 2022; $n = 27$ between August 2022 and October 2022). For the present analyses, both groups were combined. Both groups were treated with an identical data collection method as mentioned above. Two further groups served as controls: 1) “Unimpaired Post-COVID” (UPC): a collective of 38 subjects (age: 42 [24]; 13/25 males/females – self-reported) in the chronic phase following PCR-confirmed COVID-19 infection, without persistent subjective complaints and enrolled between August and October

2022. ... Although the UPC group was recruited with a delay, it took place in the same time period in which PCC patients were also included and examined (n=27). 2) “Healthy non-COVID” (HNC group): an in-house collective of 46 healthy subjects (age: 44 [31]; range: 21 to 80 years; 23/23 males/females – self-reported), with no history of COVID-19 infection (obtained from medical records and self-reports) and no significant difference in age to subjects in the PCC and UPC groups (Mann-Whitney-U, $P = 0.22$). ... HNC subjects were enrolled between June 2020 and January 2022.”

It would be great if an additional secondary analysis could be carried out, this time adding the period when each participant was acquired as a discrete confounder, on top of sex and age. To address the reviewer's concerns related to the initial submission (i.e. NCOMMS-22-15714-T), we enrolled a further control group of patients that had contracted COVID-19 without persistent subjective complaints (i.e. UPC-group) and expanded the PCC group by 27 further patients. Thus, a gap of approximately 7 months exists within the overall recruitment process. Given the overall duration of the recruitment process (6/2020 - 10/2022 for period 1), this gap of approximately 7 months to recruitment period 2 is rather small. Moreover, PCC patients, UPC- and HNC-subjects were examined on the same scanner with the same coil using identical sequences and identical clinical testing was carried out by the same practitioners. From this point of view, we do not assume that a Type I error exists.

Adding the period as a confounder on top of sex and age is however problematic from a statistical point of view: as UPC-subjects were enrolled in period 2 only and HNC-subjects were only enrolled in period 1, the covariate “period” is inextricably linked to group identities and no longer a discrete variable. Consequently, including the covariate “period” in the voxel-based group comparison of V-extra cannot be used meaningfully.

To overcome this problem and to address the reviewer's concern, we firstly performed intergroup comparison of whole-brain gray-matter DMI parameters using ANCOVAs, controlling for “age” and “sex” separately for period 1 and 2 PCC patients. Although the significance level in period 2 is lowered due to the lower number of patients (n=27), effect directions were similar with a reduction of V-extra and an increase in V-CSF when compared to HNC-controls (**Review-specific Figure 1**) and UPC-subjects (**Review-specific Figure 2**).

Review-specific Figure 1: Box plots show the distribution of microstructural compartments within the entire gray matter. In patients with Post-COVID-Condition (PCC), a decrease in the extraneurite volume fraction (V-extra) was accompanied by a significant increase in the free-fluid fraction (V-CSF) when compared to the Healthy Non-COVID (HNC) group. This effect persists when analyses were computed separately for period 1 and 2 PCC patients. Group comparisons were performed using ANCOVAs, with “age” and “sex” as nuisance covariates; * $P < 0.05$, *** $P < 0.001$.

Review-specific Figure 2: Box plots show the distribution of microstructural compartments within the entire gray matter. In patients with Post-COVID-Condition (PCC), a decrease in the extraneurite volume fraction (V-extra) was accompanied by a significant increase in the free-fluid fraction (V-CSF) when compared to the Unimpaired Post-COVID (UPC) group. This effect persists when analyses were computed separately for period 1 and 2 PCC patients. Group comparisons were performed using ANCOVAs, with “age” and “sex” as nuisance covariates; *** $P < 0.001$.

Moreover, we performed novel voxel-based comparisons of V-extra between PCC vs. HNC and PCC vs. UPC groups. Due to the small number of PCC patients in period 2, we decided against carrying out voxel-based analysis with this particular group alone. Instead, we compared PCC-patients enrolled in period 1 with the combined collective (i.e. period 1+2) to test if the inclusion of the novel patients led to any changes. However, adding the PCC-patients enrolled in period 2 did not change spatial patterns of significant effects or effect size/directions expressed by beta-coefficients for both, PCC vs. HNC (see **Review-specific Figure 3**) and PCC vs. UPC (see **Review-specific Figure 4**). In summary, the inclusion of the 27 PCC patients from the second recruitment interval actually confirmed the results of the first analysis.

Review-specific Figure 3. Top: Results of statistical voxel-wise analyses of V-extra after threshold-free cluster enhancement and family-wise-error (FWE)-correction between different groups: Post-COVID-Condition (PCC) vs. Healthy Non-COVID controls (HNC). Voxels with significantly different V-extra were indicated by shading and superimposed onto a T1w MRI template (top rows). P-values were corrected for multiple comparisons across voxels using the family-wise error rate (FWE), with “age” and “sex” as nuisance covariates. Radiological orientation: left side of the image corresponds to the patient’s right; numbers denote the axial (z) position in millimeters. P1 comprises PCC patients enrolled in period 1, P1+2 the entire collective of n = 89 patients. **Bottom:** Results of standardized regression coefficients derived from the same model as indicated above between different groups: Post-COVID-Condition (PCC) vs. Healthy Non-COVID controls (HNC). Color-coding indicates the beta coefficient values as a measure of effect size of the factor “COVID-19” (hot colors: positive effects vs. cold colors: negative effects; bottom rows). Radiological orientation: left side of the image corresponds to the patient’s right; numbers denote the axial (z) position in millimeters.

Post-COVID-Condition (PCC) vs. Unimpaired Post-COVID (UPC)

Review-specific Figure 4 Top: Results of statistical voxel-wise analyses of V-extra after threshold-free cluster enhancement and family-wise-error (FWE)-correction between different groups: Post-COVID-Condition (PCC) vs. Unimpaired Post-COVID group (UPC). Voxels with significantly different V-extra were indicated by shading and superimposed onto a T1w MRI template (top rows). P-values were corrected for multiple comparisons across voxels using the family-wise error rate (FWE), with “age” and “sex” as nuisance covariates. Radiological orientation: left side of the image corresponds to the patient’s right; numbers denote the axial (z) position in millimeters. P1 comprises PCC patients enrolled in period 1, P1+2 the entire collective of $n = 89$ patients. **Bottom:** Results of standardized regression coefficients derived from the same model as indicated above between different groups: Unimpaired Post-COVID group (UPC). Color-coding indicates the beta coefficient values as a measure of effect size of the factor “COVID-19” (hot colors: positive effects vs. cold colors: negative effects; bottom rows). Radiological orientation: left side of the image corresponds to the patient’s right; numbers denote the axial (z) position in millimeters.

Regarding voxel-based associations between V-extra and clinical parameters, inclusion of “period” as a third covariate on top of “age” and “sex” came at the expense of statistically significant voxels after FWE correction for multiple comparisons. In general, it is well known that the addition of covariates lowers the overall level of significance. However, to investigate whether the pattern of effects was affected by inclusion of “period”, we compared beta coefficient-maps of voxel-based analyses with V-extra as the dependent variable after TFCE and FWE correction for both covariates “age” + “sex” vs. “period” + “age” + “sex”. Here, effect-patterns were consistent between models for MoCA, Olfaction and WEIMuS (see **Review-specific Figure 5**). Thus, our finding of symptom-specificity of affected networks was confirmed despite the addition of “period” as covariate.

Cognitive performance (MoCA)

TFCE + FWE ($p < 0.05$), Cov. „age“ + „sex“

TFCE + FWE ($p < 0.05$), Cov. „period“ + „age“ + „sex“

Olfaction

TFCE + FWE ($p < 0.05$), Cov. „age“ + „sex“

TFCE + FWE ($p < 0.05$), Cov. „period“ + „age“ + „sex“

Fatigue (WEIMuS)

TFCE + FWE ($p < 0.05$), Cov. „age“ + „sex“

TFCE + FWE ($p < 0.05$), Cov. „period“ + „age“ + „sex“

Review-specific Figure 5. Results of standardized regression coefficients derived from voxel-based analyses of associations between clinical scores and V-extra as the dependent variable after TFCE and FWE correction for multiple comparisons. The model was computed with covariates “age”+ “sex” (upper panel) and “period” + “age” + “sex” (lower panel). The addition of “period” did not influence the characteristic orientations of the symptom-specific patterns. Color-coding indicates the beta coefficient values as a measure of the effect size of the factor “COVID-19” (hot colors: positive effects vs. cold colors: negative effects; bottom rows). Radiological orientation: left side of the image corresponds to the patient’s right; numbers denote the axial (z) position in millimeters.

Reviewer #3 (Remarks to the Author):

The authors satisfactorily addressed reviewers' comments, I have nothing further to suggest. The manuscript can be published in its present form.

We thank the reviewer for his positive evaluation of our effort!

***SPECIFIC REQUEST:**

You will see from the Methods section that the PCC sample was recruited in two separate time periods and then combined. It would be helpful to us if you could also comment specifically on whether you have any concerns with this decision, especially in relation to possible inflation of Type 1 errors.

I am not worried about ~7 month lapse in recruitment of PCC sample between January and August 2022, given the overall duration of the recruitment process (6/2020 - 10/2022). Authors also addressed a related concern #4 of Reviewer 1 about delay between infection and MRI.

REVIEWERS' COMMENTS

Reviewer #1 (Remarks to the Author):

The authors have very comprehensively carried out ancillary analyses to allay any remaining concerns the Editor may have had about the 7-month gap in the recruitment of some of the participants (my suggestion was indeed impossible to implement as is).

The newly created review-specific figures should be added as Supplementary Material, and a (very short) paragraph summarising the outcomes of these analyses added to the Discussion.

Authors' Response to Reviewers Comments:

Manuscript Number: NCOMMS-22-15714D

MS TITLE: Cerebral microstructural alterations in Post-COVID-condition are related to cognitive impairment, olfactory dysfunction and fatigue

We again thank the reviewer for the thorough review and valuable comments on our manuscript. In the following, we provide a point-by-point reply describing how we addressed the individual comments (answers are reported in red, and changes to the manuscript in *italic red*).

Reviewer #1 (Remarks to the Author):

The authors have very comprehensively carried out ancillary analyses to allay any remaining concerns the Editor may have had about the 7-month gap in the recruitment of some of the participants (my suggestion was indeed impossible to implement as is). The newly created review-specific figures should be added as Supplementary Material, and a (very short) paragraph summarizing the outcomes of these analyses added to the Discussion.

We appreciate this positive feedback!

As the reviewer suggests, we included the Figures into the Supplementary Information (i.e. Supplementary Figures 1, 2, 5, 6, and 8) and implemented a short summarizing paragraph into the Discussion. Moreover, at the request of the editorial office, we have also incorporated the analyses into the Results section of the revised manuscript. In summary, we implemented the following changes:

Results section, page 9: *“Since the patients with Post-COVID-Condition were recruited in two periods, we conducted the comparison of whole-brain gray-matter diffusion MRI parameters separately for both periods. Here, comparison with HNC- (Supplementary Figure 1) and UPC participants (Supplementary Figure 2) were similar to the pooled analysis.”*

Results section, page 11: *“Regarding the two recruitment periods of patients with Post-COVID-Condition (PCC), we performed voxel-based comparisons of V-extra of patients enrolled in period 1 with the combined collective (i.e. period 1 and 2). Here, adding the PCC-patients enrolled in*

period 2 did not change spatial patterns of significant effects or effect size/directions expressed by beta-coefficients for both, PCC vs. HNC (Supplementary Figure 5) and PCC vs. UPC (Supplementary Figure 6)."

Result section, page 12: *"To account for the two recruitment periods of patients with Post-COVID-Condition (PCC), we furthermore added the period as third nuisance covariate in addition to age and sex into our above mentioned models. However, effect-patterns were consistent between models for cognition, olfaction and fatigue, confirming the symptom-specificity of affected networks (Supplementary Figure 8)."*

Discussion (Limitations section), page 19/20: *"It must also be noted that a gap of 7 months divided the recruitment of PCC patients into two periods. However, the period had no influence on the changes in gray matter diffusion MRI parameters (Supplementary Figures 1 and 2), the spatial distribution and direction of V-extra changes (Supplementary Figures 5 and 6), as well as the symptom-specific reduction of V-extra (Supplementary Figure 8)."*